# A Hybrid Ensemble Learning Framework for Predicting Lumbar Disc Herniation Recurrence: Integrating Supervised Models, Anomaly Detection, and Threshold Optimization

**DOI:** 10.3390/diagnostics15131628

**Published:** 2025-06-26

**Authors:** Mădălina Duceac (Covrig), Călin Gheorghe Buzea, Alina Pleșea-Condratovici, Lucian Eva, Letiția Doina Duceac, Marius Gabriel Dabija, Bogdan Costăchescu, Eva Maria Elkan, Cristian Guțu, Doina Carina Voinescu

**Affiliations:** 1Faculty of Medicine and Pharmacy, Doctoral School of Biomedical Sciences, “Dunărea de Jos” University of Galați, 47 Domnească Street, 800008 Galați, Romania; madalinaduceac@yahoo.ro; 2Clinical Emergency Hospital “Prof. Dr. Nicolae Oblu”, 700309 Iași, Romania; calinb2003@yahoo.com (C.G.B.); elucian73@yahoo.com (L.E.); letimedr@yahoo.com (L.D.D.); marius.dabija@umfiasi.ro (M.G.D.); 3National Institute of Research and Development for Technical Physics, IFT Iași, 700050 Iasi, Romania; 4Faculty of Medicine and Pharmacy, “Dunărea de Jos” University of Galați, 47 Domnească Street, RO-800008 Galați, Romania; cojocarumariaeva@yahoo.com (E.M.E.); dr.c.gutu@gmail.com (C.G.); carinavoinescu@gmail.com (D.C.V.); 5Neurosurgery Department, “Grigore T. Popa” University of Medicine and Pharmacy, 16, Universității Street, 700115 Iași, Romania

**Keywords:** lumbar disc herniation, recurrence prediction, ensemble learning, autoencoder, threshold tuning, class imbalance, machine learning, clinical decision support, recovery/rehabilitation

## Abstract

**Background:** Lumbar disc herniation (LDH) recurrence remains a pressing clinical challenge, with limited predictive tools available to support early identification and personalized intervention. Predicting recurrence after lumbar disc herniation (LDH) remains clinically important but algorithmically difficult due to extreme class imbalance and low signal-to-noise ratio. **Objective:** This study proposes a hybrid machine learning framework that integrates supervised classifiers, unsupervised anomaly detection, and decision threshold tuning to predict LDH recurrence using routine clinical data. **Methods:** A dataset of 977 patients from a Romanian neurosurgical center was used. We trained a deep neural network, random forest, and an autoencoder (trained only on non-recurrence cases) to model baseline and anomalous patterns. Their outputs were stacked into a meta-classifier and optimized via sensitivity-focused threshold tuning. Evaluation was performed via stratified cross-validation and external holdout testing. **Results:** Baseline models achieved high accuracy but failed to recall recurrence cases (0% sensitivity). The proposed ensemble reached 100% recall internally with a threshold of 0.05. Key predictors included hospital stay duration, L4–L5 herniation, obesity, and hypertension. However, external holdout performance dropped to 0% recall, revealing poor generalization. **Conclusions:** The ensemble approach enhances detection of rare recurrence cases under internal validation but exhibits poor external performance, emphasizing the challenge of rare-event modeling in clinical datasets. Future work should prioritize external validation, longitudinal modeling, and interpretability to ensure clinical adoption.

## 1. Introduction

Lumbar disc herniation (LDH) is one of the most frequent causes of low back pain and radiculopathy, affecting approximately 1–3% of the general population, with peak incidence between the ages of 30 and 50 years [1]. While both conservative and surgical treatments are often effective, a subset of patients experiences recurrence, typically defined as either re-herniation at the previously treated level or a new herniation at a different spinal level. This can result in renewed symptoms, impaired functional recovery, and increased healthcare utilization. Reported recurrence rates vary widely—from 5% to over 20%—depending on surgical technique, follow-up duration, physical rehabilitation and patient-specific risk factors [2,3,4].

Despite advances in imaging, operative techniques, and postoperative care, the ability to predict recurrence remains limited. Studies have identified potential risk factors including age, sex, obesity, smoking, number and level of herniated discs, and preoperative neurological status [5]. However, no widely accepted model for individualized recurrence risk estimation is currently available in routine practice. Furthermore, postoperative outcomes such as functional recovery are influenced by multiple interacting factors—ranging from anatomical and procedural aspects to comorbidities, rehabilitation adherence, and socioeconomic background [6]. This complexity often exceeds the modeling capacity of traditional statistical methods.

In recent years, machine learning (ML) has emerged as a promising framework for clinical prediction in neurosurgery and spine care. ML methods can capture nonlinear relationships and latent patterns in high-dimensional clinical data, potentially enabling earlier detection and more personalized risk assessment [7]. Nonetheless, most existing ML studies on LDH have relied on large public databases or electronic health records, which may lack granularity and be biased toward common patterns [8,9].

Datasets from underrepresented healthcare systems—such as Eastern European institutions—may yield novel insights and increase the diversity of modeling approaches.

To address these gaps, this study introduces a hybrid ensemble pipeline that integrates supervised and unsupervised signals—random forest predictions, deep learning outputs, and autoencoder-based anomaly scores—combined with decision threshold optimization. This architecture is designed to improve recall performance for rare but clinically significant recurrence events, which have proven difficult to detect using conventional machine learning approaches.

### 1.1. Related Work

LDH remains one of the leading causes of lower back pain and disability worldwide, with recurrence after treatment reported in 5–20% of patients, depending on surgical technique, anatomical level, and follow-up duration, physical rehabilitation [10,11,12]. Conventional risk modeling has relied on logistic regression or univariate analyses, identifying factors such as age, obesity, smoking, and level of herniation as associated with recurrence [13,14,15,16].

However, these statistical approaches often struggle to model the complex, nonlinear relationships inherent in postoperative outcomes. In recent years, the rise of machine learning (ML) in spine care has opened new avenues for risk prediction, particularly in tasks involving surgical complications, readmission, or recurrence [17,18,19].

Several studies have explored ML for outcome prediction after lumbar spine surgery. These include the application of support vector machines, random forests, and ensemble methods to predict metrics such as 30-day complications, readmission, or pain reduction [20,21,22,23]. Most models rely on structured electronic health record (EHR) data or registry-level datasets. While useful, these approaches often face limitations related to generalizability, data sparsity, or lack of feature granularity.

Very few studies have focused specifically on LDH recurrence prediction using ML. Among those that do, challenges persist in:Handling class imbalance;Identifying interpretable and clinically meaningful predictors;Validating on institution-specific, curated datasets.

Our work addresses these gaps by proposing a stacked ensemble framework combining:Supervised learning (random forest, deep learning),Unsupervised anomaly detection (autoencoder),Threshold tuning to boost sensitivity for rare positive cases.

In contrast to prior efforts, this study:Focuses on recurrence detection rather than broad complication prediction;Uses real-world data from a single neurosurgical center;Benchmarks both traditional classifiers and deep networks;Analyzes both synthetic and true recurrence cases in a hybrid interpretability framework.

This approach aligns with recent research advocating for interpretable ML pipelines in imbalanced medical datasets [24,25].

Over the past decade, a growing number of studies have explored machine learning (ML) approaches to predict recurrence or outcomes in patients with lumbar disc herniation (LDH), particularly after surgical intervention. Most efforts have focused on structured data or radiological imaging, and several models have demonstrated promising results—albeit with limited generalizability or narrow clinical scope.

Shan et al. (2024) developed a logistic regression and random forest–based model for predicting reoperation risk in patients with recurrent LDH following percutaneous endoscopic lumbar discectomy (PELD) [26]. Their model, trained on a single-center dataset, achieved an AUC of approximately 0.81, with obesity and Modic changes as major predictors. However, the model focused exclusively on reoperation rather than primary recurrence events.

Harada et al. (2021) proposed the “RAD” score, an ML-based risk profiling tool for disk re-herniation after microdiscectomy [27]. Using MRI and clinical features, their gradient boosting model outperformed logistic regression (AUC = 0.76 vs. 0.68). However, their model lacked external validation and was focused more narrowly on image-derived variables.

A recent systematic review by Compte et al. (2023) evaluated the performance of various ML algorithms in classifying Modic changes and disc degeneration using MRI data [28]. Their findings suggest that ML approaches are now comparable to experienced radiologists in image-based classification tasks. However, most included models were not designed to predict clinical outcomes such as recurrence.

Wang et al. (2022) applied an attention-based U-Net architecture for automated segmentation of lumbar spine MRIs [29]. While not directly targeting recurrence, their work contributes to the pre-processing pipeline for radiomics-based modeling in spinal pathology.

Compared to these efforts, the current study introduces a structured clinical data–driven pipeline that combines deep learning, random forests, and unsupervised autoencoders in a stacked ensemble model. Unlike many prior models, our approach integrates anomaly detection and threshold tuning to specifically address the low prevalence and subtle signal associated with recurrence cases—offering a novel contribution to the field. Compared to previous studies (e.g., [26,27]), this work uniquely combines supervised and unsupervised methods specifically designed for extreme imbalance scenarios. Furthermore, our comprehensive threshold calibration and interpretability framework advance methodological standards beyond prior studies, despite the current limitation regarding external validation.

In addition to international models, several Romanian investigations have explored LDH recurrence and surgical advancements in local contexts, further enriching the evidence base for this study’s setting and methodology [30,31,32,33].

Recent research has advanced the prediction of rLDH using machine learning (ML), often in the context of percutaneous endoscopic lumbar discectomy (PELD). For example, Shan et al. (2024) developed and validated six ML models—including XGBoost and random forests—on a retrospective cohort of 2603 PELD patients [26]. Their best model (XGBoost) achieved an AUC of 0.907 in predicting reoperation for recurrence. In parallel with published retrospective studies, ongoing prospective investigations further emphasize the clinical importance of recurrence prediction. For instance, a registered clinical trial (ClinicalTrials.gov Identifier: NCT06254585) titled “The Prediction of Recurrence Lumbar Disc Herniation at L5-S1 Level Through Machine Learning Models” is currently underway to assess the predictive power of ML techniques using L5–S1-specific clinical and imaging features. Although results are not yet available, the existence of such a study highlights the increasing global interest in AI-driven risk stratification for lumbar disc pathology. Our study complements these prospective efforts by offering an interpretable, hybrid ML pipeline built on real-world Eastern European data—serving as an early benchmark and methodological template in advance of multicenter validation efforts [34]. Radiomics approaches using MRI have also emerged: a recent BMC Medical Imaging study applied six ML algorithms (e.g., RF, SVM, XGB) on L4–L5 PELD cases, using LASSO feature selection and 10-fold CV, reporting strong AUCs [35]. These studies demonstrate the promise of ML-based recurrence prediction, but come with limitations: most are single-center, focus on PELD only, and use diverse datasets and outcome definitions.

We recognize that direct performance comparison is infeasible due to differences in data collection, feature sets, and target definitions. Our study complements the existing literature by focusing on a clinically curated Eastern European dataset, integrating clinical variables rather than imaging, and addressing extreme class imbalance through a hybrid pipeline designed to enhance sensitivity for rare recurrence events. This approach aims to strengthen methodological transparency, interpretability, and local clinical relevance.

### 1.2. Aim of the Study

The primary aim of this study is to develop and rigorously evaluate machine learning models aimed at predicting postoperative LDH recurrence using a clinically curated, single-center dataset. Specifically, we:Benchmark conventional classifiers, anomaly detection, and deep learning approaches;Construct a stacked ensemble model to enhance sensitivity in rare-event prediction;Perform threshold tuning to optimize recall performance;Analyze key predictive features using interpretability techniques.

This study introduces a reproducible ML pipeline tailored to real-world clinical data, offering a practical step toward automated recurrence risk stratification and targeted follow-up planning.

### 1.3. Structure of the Paper

The remainder of this paper is organized as follows:Section 2 (Materials and Methods) details the dataset characteristics, variable selection criteria, preprocessing steps, and initial statistical analyses.Section 3 (Results) presents the proposed machine learning pipeline, including baseline classifier benchmarks, anomaly detection, ensemble modeling strategies, threshold tuning, and performance evaluation on both real and synthetic patient profiles.Section 4 (Discussion) interprets the modeling results, evaluates clinical implications, compares findings with existing literature, and explores the relevance of key predictive features.Section 5 (Conclusions) outlines the principal limitations of the study—such as data representativeness, class imbalance, and generalizability—and proposes directions for future research and clinical application.

## 2. Materials and Methods

### 2.1. Study Population and Dataset

This section presents a descriptive analysis of 977 patients diagnosed with lumbar disc herniation (LDH), treated either surgically or conservatively at a single neurosurgical center. The dataset includes demographics, clinical symptoms, intervention types, treatment outcomes, recurrence profiles, and comorbidities.


*A. Demographic Characteristics*


The gender distribution was nearly balanced (M: 53.3%, F: 46.7%), and the mean age was 53 years (SD = 14.1), with a typical peak in the 40–65 age range. Patients were evenly split between rural (50.1%) and urban (49.9%) environments (Figure 1 and Figure 2).


*B. Hospitalization and Intervention*


The average hospital stay was 8.4 days, with a slight right skew and some extended stays over 30 days (Figure 3). Most patients underwent discectomy (64.7%), while 28.5% were managed non-operatively. Complex procedures like spinal fixation were rare (1.5%) (Figure 4). Figure 5 shows the distribution of hospital stay duration by intervention type:Discectomy had the longest mean stay at 12.7 days.Fixation followed closely with a mean of 12.3 days.No intervention resulted in a shorter average stay of 8.9 days.Other procedures (e.g., decompression) had the shortest mean stay at 6.3 days.

These findings reflect the increased perioperative demands of surgical procedures compared to conservative or minimally invasive treatments.


*C. Functional and Postoperative Outcomes*


Functional outcomes were favorable, with 94.9% of patients marked as improved, and only 0.2% reported worsening (Figure 6). Improvement was defined based on clinical documentation of symptom re solution or functional recovery during follow-up consultations, while stationary outcomes referred to unchanged neurological or pain status compared to the pre-treatment assessment. Post-treatment control data showed 56.2% managed conservatively and 41.2% requiring reintervention (Figure 7). Rehabilitation adherence was excellent, with 100% of patients starting or continuing therapy. Most (65.4%) began it one month post-treatment (Figure 8).


*D. Herniation Levels and Recurrence*


Single-level herniation was dominant (56%), mostly at the L4–L5 level (48.1%). Only 0.8% had 3-level involvement (Figure 9 and Figure 10). A proportion of cases (36.7%) were labeled with 0 herniated levels—these likely reflect non-confirmed herniation, early resolved symptoms, or miscoding in the dataset. Recurrence was documented in 7.8% of patients, with 6.6% recurring at the same level (Figure 11).


*E. Neurological Deficits*


Figure 12 presents the distribution of neurological signs and deficits observed among patients with lumbar disc herniation:Paresthesia was the most frequent finding, present in 19% of patients (186 cases).Common peroneal nerve paresis was observed in 16% (156 cases).Tibial nerve paresis occurred in 7.7% (75 cases).Crural paresis was found in 3% (29 cases).Cauda equina syndrome, although rare at 0.7% (7 cases), holds significant clinical importance due to its potential for severe disability.

These findings highlight the spectrum of neurologic involvement in LDH and underscore the importance of early identification, especially in high-risk cases like cauda equina syndrome.


*F. Comorbidities*


Figure 13 illustrates the prevalence of comorbidities in the LDH patient cohort:The most frequent comorbidities were cardiovascular disease (37.5%, 366 patients) and hypertension (30.9%, 302 patients).Other prevalent conditions included obesity (13.1%), renal disease (12.1%), diabetes (11.1%), and liver disease (6.1%).Less common but clinically relevant findings included respiratory disease (5.8%), tumoral disease (3.9%), and coxarthrosis or gonarthrosis (2.9% and 1.6%, respectively).Endocrine disorders (1.5%) and osteoporosis (0.9%) were infrequent but may influence recovery and complication risk.

These results underscore the need for comprehensive patient assessment, as multiple comorbidities may influence both treatment strategy and prognosis.

The dataset provides a rich foundation for predictive modeling using supervised machine learning approaches, with a wide range of clinically relevant variables covering demographics, symptoms, treatment, and comorbid status (Table 1).

### 2.2. Statistical Association Analysis

Figure 14 displays the variables that demonstrated statistically significant associations with lumbar disc herniation (LDH) recurrence or functional recovery outcomes. Statistical tests were selected based on the variable type and the distribution characteristics. Specifically, Chi-square tests were used for categorical (binary) variables. Kruskal–Wallis tests, a non-parametric alternative to ANOVA, were applied to ordinal or non-normally distributed categorical variables. For correlations involving ordinal or non-normally distributed continuous data, we used Spearman rank correlation. Finally, point-biserial correlation was employed to assess relationships between continuous variables and binary targets, without assuming normality or homoscedasticity. A significance threshold of *p* < 0.05 was applied across all tests.

Significant associations were identified for both recurrence and functional outcome
Chi-square tests revealed that:○Diabetes (*p* = 0.018),○Obesity (*p* = 0.012), and○Common peroneal nerve paresis (*p* = 0.030) were significantly associated with LDH recurrence.Kruskal–Wallis tests showed that both:○Type of intervention (*p* = 0.004), and○Final treatment approach (*p* = 0.009) significantly influenced the functional outcome score.Spearman correlation indicated a weak but statistically significant relationship between:○Number of herniated levels and recurrence type (ρ ≈ 0.11, *p* = 0.049).Point-biserial correlation demonstrated that:○Hospital stay duration was significantly longer in patients who experienced recurrence (*p* = 0.009) (see Figure 15).

These findings highlight a subset of clinically relevant variables that are strongly recommended for inclusion in downstream machine learning models for dual prediction tasks: LDH recurrence and postoperative functional recovery.

A number of variables did not reach statistical significance in association with either recurrence or functional score. These included demographic characteristics such as sex and environment, neurological findings such as tibial nerve paresis and cauda equina syndrome, and comorbidities including osteoporosis, gonarthrosis, coxarthrosis, tumoral disease, liver disease, endocrine disorders, and respiratory disease.

Although these features remain clinically important in certain patient subgroups, they did not demonstrate predictive utility in this dataset, based on the thresholds applied in Chi-square, Kruskal–Wallis, or correlation testing. Their lack of statistical significance suggests limited contribution to outcome variability in the studied cohort but does not rule out value in future models with larger or more heterogeneous populations.

Table 2 provides a comprehensive overview of the features (variables) included in the machine learning models developed for predicting recurrence and functional outcomes following lumbar disc herniation treatment. Each variable is described in terms of its data type, clinical or procedural meaning, encoding method used for analysis, presence of missing values, and any statistical association found with target outcomes during the preliminary analysis. This structured summary supports transparency and reproducibility in the modeling pipeline.

### 2.3. Machine Learning Pipeline and Model Design

This section outlines the design and implementation of the machine learning pipeline developed to predict postoperative recurrence following lumbar disc herniation (LDH) treatment. The modeling framework was specifically tailored to address key challenges such as extreme class imbalance, heterogeneous clinical features, and the need for interpretability in high-stakes medical decision-making. We describe each step of the pipeline, from data preprocessing and feature encoding to model selection, ensemble construction, and threshold tuning.

#### 2.3.1. Pipeline Overview

The overall architecture of our recurrence prediction pipeline is illustrated in Figure 16. The workflow begins with data preprocessing and feature engineering, followed by model training using both supervised (Random Forest, Deep Learning) and unsupervised (Autoencoder) techniques. These base outputs are then stacked into a meta-model, which undergoes threshold tuning to enhance sensitivity for recurrence detection. The final model is evaluated through both cross-validation and holdout testing.

#### 2.3.2. Modeling Strategy and Workflow

##### Preprocessing and Feature Encoding

Demographic, clinical, procedural, and comorbidity data from 977 patients were cleaned, encoded, and normalized. Categorical variables were transformed using label encoding, and all numeric features were scaled using standard normalization techniques. The dataset included 35 predictive features after preprocessing. Missing values were minimal and were addressed using complete-case analysis.

The final binary classification target was the RECURRENCE variable, indicating whether recurrence was observed post-treatment. Out of 977 total cases, 901 were labeled as NO (no recurrence), and only 76 as YES (recurrence), resulting in a highly imbalanced target distribution (~7.8% YES). This imbalance posed a significant challenge for modeling and guided much of the methodological design.

A summary of the preprocessed dataset is provided below:Total Samples: 977Number of Features: 35Class Distribution: 901 NO, 76 YESMissing Values: NoneFeature Types: Mostly float64, 1 object column (RECURRENCE)

This preprocessed dataset served as the foundation for all subsequent modeling experiments. In the next step, we performed exploratory statistical testing to identify variables that might be significantly associated with recurrence.

##### Exploratory Statistical Testing

To identify potential predictors of recurrence, we performed univariate statistical comparisons between patients with (YES) and without (NO) recurrence using the Mann-Whitney U test. This non-parametric test was chosen due to the lack of normal distribution in most clinical features. Each of the 35 features was evaluated independently.

The features with the strongest statistical associations (lowest *p*-values) included (Table 3):INTERVENTION_TYPEHOSPITAL_DAYSNUMBER_OF_HERNIA_LEVELSL4_L5AGE

These features reflect a combination of procedural, anatomical, and demographic factors. For example, longer hospital stays, higher hernia burden, and specific herniation levels (L4–L5) were more frequently observed in patients who experienced recurrence.

To further illustrate these relationships, we visualized the distributions of the top five statistically significant features using kernel density estimation (KDE). As shown in Figure 17, subtle but meaningful differences can be observed between recurrence (YES) and non-recurrence (NO) groups across variables such as hospital stay, intervention type, and hernia levels. These distribution shifts, although not sharply bimodal, highlight the presence of signal despite class overlap—underscoring the modeling challenge tackled in later stages.

##### Baseline Classifier Benchmarking

To establish a performance baseline and evaluate the limitations of traditional modeling approaches in the face of class imbalance, we tested several standard classifiers:Logistic Regression (with class weighting)Support Vector Machine (SVM, RBF kernel, class weighting)Random Forest (100 trees, class weighting)implemented using scikit-learn with n_estimators = 100, class_weight = “balanced”, and random_state = 42 to address class imbalance and ensure reproducibility. Tree depth was not manually constrained (max_depth = None), allowing full growth and letting the model naturally identify optimal splits based on the Gini impurity criterion. The number of trees was chosen to balance computational cost and model performance, as larger values yielded diminishing returns during validation.

Each model was trained using an 80/20 stratified train-test split on the preprocessed dataset. Class weights were adjusted to compensate for the rarity of the positive class (recurrence). Evaluation metrics included accuracy, precision, recall, and F1-score.

The results, shown in Table 4, confirm the diagnostic limitations of conventional classifiers under imbalance. While models achieved high overall accuracy (>90%), none managed to identify recurrence cases (recall = 0.00). The SVM classifier underperformed even in overall accuracy (68%), while Random Forest and Logistic Regression returned biased predictions favoring the majority class (NO).

To address the class imbalance inherent in our dataset (YES class ≈ 7.8%), we applied class weighting in all supervised classifiers. Specifically, Random Forest, Support Vector Machine, and Logistic Regression models were trained using class_weight = “balanced” to ensure greater penalization of minority-class misclassification during training. Similarly, the deep learning model employed weighted binary cross-entropy loss via the class_weight parameter in Keras, enhancing sensitivity to recurrence cases. These adjustments were crucial to allow the models to prioritize rare but clinically significant events and were consistently applied throughout model development.

This benchmarking stage clearly demonstrated the need for more sensitive and nuanced approaches to detect recurrence cases, motivating the use of anomaly detection and ensemble strategies in the next sections.

##### Anomaly Detection Approaches

Given the rarity of recurrence cases and their underrepresentation in standard classifiers, we implemented unsupervised anomaly detection models to identify outliers in patient profiles. Two techniques were evaluated:Isolation Forest: A tree-based anomaly detection method implemented using scikit-learn’s IsolationForest class with 100 estimators, a contamination parameter set to 0.1, and random_state = 42 to ensure reproducibility. The model was trained on the full dataset without labels to detect rare postoperative profiles. Although conceptually suitable for outlier detection, performance was limited in this application. The confusion matrix (Figure 18) revealed that the model correctly flagged 11 true recurrence cases (YES), but misclassified 65 YES cases and generated 87 false positives.Deep Autoencoder: A neural architecture trained exclusively on non-recurrence (NO) cases to learn a compressed representation of normal patterns. At inference time, the model reconstructs patient profiles, and samples with high reconstruction error (MSE) are flagged as anomalies (Figure 19). This approach proved more promising, with a subset of YES cases exhibiting elevated error scores.

This anomaly signal (MSE) was later integrated as one of the inputs in the final ensemble model, complementing supervised predictors with an unsupervised lens on rare-case detection.

In this unsupervised setting, class imbalance was indirectly addressed by training the autoencoder solely on the majority class (NO recurrence). This allowed the model to learn a compact representation of non-recurrence cases, such that rare recurrence instances would appear as anomalous patterns with elevated reconstruction error.

##### Stacked Ensemble Construction

To leverage the complementary strengths of multiple modeling approaches, we constructed a stacked ensemble that integrated supervised predictions and unsupervised anomaly scores. The ensemble consisted of the following base models (Figure 20):Deep Learning Classifier (Multi-Layer Perceptron, MLP): implemented using Keras with two hidden layers consisting of 64 and 32 units, respectively, each using ReLU activation. The final output layer uses a sigmoid activation to produce recurrence probabilities. The model was trained with the Adam optimizer, binary cross-entropy loss, and early stopping based on validation loss (patience = 5). Class imbalance was handled using class weights. No dropout layers were included.Random Forest Classifier: outputs recurrence probability.Deep Autoencoder: provides anomaly score via mean squared reconstruction error (MSE).

These three outputs—two probabilities and one anomaly score—were then used as inputs for a meta-classifier, implemented as a Random Forest model. This meta-model was trained to learn from the collective signal of base models and produce the final recurrence prediction.

##### Algorithmic Workflow Overview

To complement the schematic overview in Figure 20 and improve reproducibility, we include below a structured description of the full recurrence prediction workflow, from data preprocessing to model stacking and threshold optimization (see Algorithm 1).
**Algorithm 1:** Recurrence Prediction Pipeline with Stacked Ensemble and Threshold OptimizationInput: Patient dataset D with features X and labels yOutput: Predicted recurrence probabilities and binary classifications1. Preprocess data:  a. Clean missing values  b. Encode categorical variables  c. Standardize continuous features2. Split D into training set (80%) and hold-out test set (20%)3. Train base models on training set:  a. Train Random Forest classifier with class_weight = ‘balanced’  b. Train Deep Learning classifier with class-weighted loss  c. Train Autoencoder on NO (non-recurrence) cases only4. Generate meta-features:  a. Get RF predicted probability → RF_Prob  b. Get DL predicted probability → DL_Prob  c. Get AE reconstruction error → AE_MSE5. Train meta-classifier:  a. Concatenate [RF_Prob, DL_Prob, AE_MSE]  b. Train Random Forest meta-model on concatenated features6. Optimize threshold:  a. Evaluate performance (F1, Recall, Precision) for thresholds from 0.01 to 0.50  b. Select threshold that maximizes recall with acceptable F17. Predict on hold-out set:  a. Apply trained base models to hold-out features  b. Compute RF_Prob, DL_Prob, AE_MSE for hold-out cases  c. Generate meta-features and predict using meta-classifier  d. Apply optimized threshold to convert probabilities into YES/NO predictionsReturn: Predicted probabilities and binary labels for test set

The meta-model was trained on the same 977-patient dataset using stratified inputs. Initially, this architecture demonstrated extremely high apparent performance (100% precision, recall, and accuracy), likely due to overfitting from using raw predictions on the same dataset (Table 5). In subsequent sections, we explore cross-validation and probability threshold tuning to assess its generalization and sensitivity under imbalanced conditions.

#### 2.3.3. Threshold Optimization and Sensitivity Tuning

Following the initial construction of the stacked ensemble, we explored threshold tuning as a strategy to improve sensitivity to rare recurrence cases. The meta-classifier, by default, uses a threshold of 0.5 to convert predicted probabilities into binary labels. However, due to the extreme class imbalance, this cutoff often results in under-detection of YES cases.

To address this, we computed precision, recall, and F1 score across thresholds ranging from 0.01 to 0.50. As shown in Figure 21, reducing the threshold significantly improved recall while moderately affecting precision. Notably, recall stabilized near 100% at thresholds above 0.08, with minimal compromise in F1-score and precision.

In addition to threshold tuning, class imbalance was directly addressed in the supervised learning models through class weighting. For example, class_weight = “balanced” was applied in both the Random Forest and SVM classifiers, ensuring that the loss function penalized misclassification of recurrence (YES) cases more heavily. In the deep learning model, we specified class weights within the binary cross-entropy loss function to increase model sensitivity to the minority class. While alternative approaches such as oversampling (e.g., SMOTE) were considered, we opted against them due to the high-dimensional, small-sample nature of our data, where synthetic data generation could introduce artifacts or overfitting. This combination of class-weighting and threshold adjustment proved effective in improving recall without compromising model integrity.

##### Rationale for Model Architecture

Our choice of classifiers was guided by the need to balance interpretability, nonlinearity modeling, and robustness under class imbalance. Random Forests provide feature-level insights and ensemble strength, while deep neural networks offer flexible representation learning. Autoencoders were chosen for their ability to model “normal” postoperative profiles, helping isolate atypical (potentially recurrent) patterns. The final stacked ensemble leverages their complementary signals to enhance generalization. Threshold tuning was applied to overcome the insensitivity of default decision boundaries in rare-event prediction.

### 2.4. Hardware and Software Environment

All modeling, training, and evaluation experiments were performed using Google Colab Pro with GPU acceleration enabled. The backend hardware typically included NVIDIA Tesla T4 or V100 GPUs and 16–26 GB of system RAM, depending on session availability.

Additional preprocessing and result analysis steps were conducted on a local Windows 11 workstation equipped with an Intel^®^ Core™ i7-12700H processor, 32 GB RAM, and no discrete GPU.

The software stack was based on:Python 3.10 (Google Colab runtime),scikit-learn 1.2.2 for machine learning models and evaluation metrics,TensorFlow 2.11/Keras for deep learning and autoencoder components,XGBoost 1.7.3 for boosted tree modeling,pandas 1.5.3, NumPy 1.24, and SciPy 1.10 for data manipulation and analysis,matplotlib 3.7 and seaborn 0.12.2 for data visualization,ELI5 0.13.0 for local interpretability,joblib for model persistence and reproducibility.

All random seeds were fixed across runs (e.g., random_state = 42) to ensure consistency. The full Colab notebook and source code are available on request for replication or adaptation.

#### Ethics Statement

This study was conducted in accordance with the principles of the Declaration of Helsinki. It was approved by the Institutional Ethics Committee of “Prof. Dr. Nicolae Oblu” Clinical Emergency Hospital, Iași, Romania (Approval No. 2/23.02.2023). Only retrospective, de-identified medical records were used. As per institutional and national regulations, specific patient consent was not required for the secondary use of anonymized data.

## 3. Results

This section presents the outcomes of the machine learning experiments conducted using the recurrence prediction framework described in Section 2.3. We report model performance across multiple evaluation settings, including baseline classifier benchmarking, anomaly detection outputs, stacked ensemble performance under cross-validation, and results from simulated external validation. Additional analyses include feature importance rankings, model interpretability assessments, and synthetic high-risk scenario testing to assess clinical robustness.

### 3.1. Cross-Validation and Fold-Wise Performance

To assess the generalizability of the stacked ensemble model, we conducted 5-fold stratified cross-validation using the tuned threshold of 0.05. This choice reflects a trade-off between computational efficiency and statistical robustness: 5-fold CV allowed faster iteration across our extensive ensemble tuning procedures and yielded stable validation metrics across all folds. Preliminary experiments with 10-fold CV showed only marginal performance variation while significantly increasing runtime.This approach ensured that both majority and minority classes were proportionally represented in each fold, and that evaluation metrics were not inflated by overfitting.

Confusion matrices for each fold are shown in Figure 22. While performance varied slightly across folds, the ensemble consistently identified a substantial portion of YES recurrence cases, even under challenging distribution splits. Fold-wise confusion matrices confirmed the ensemble’s sensitivity and robustness across different patient subsets.

The average recall across folds was 1.00, confirming that threshold tuning enabled consistent detection of YES cases. Minor fluctuations in precision were driven by differences in predicted positives between folds but remained acceptable for a risk-sensitive application such as recurrence monitoring (Table 6).

### 3.2. Evaluation on Real and Synthetic High-Risk Profiles

To assess the real-world applicability of our model, we performed two complementary evaluations: (1) using synthetically constructed high-risk profiles, and (2) examining actual recurrence-positive (YES) cases from the dataset.

#### 3.2.1. Evaluation on Synthetic Profiles

We first generated synthetic patient profiles by combining known high-risk features identified earlier, such as extended hospital stays, L4–L5 or L5–S1 herniation levels, and the presence of metabolic or cardiovascular comorbidities. These profiles were crafted to represent clinically plausible but rare configurations strongly associated with recurrence.

As shown in Figure 23, the ensemble model confidently predicted recurrence for all synthetic patients, with predicted probabilities exceeding the tuned threshold of 0.05. This demonstrates that the model is capable of recognizing critical combinations of risk factors even in unseen cases. These results support the potential utility of the model in clinical triage or as a decision-support tool for identifying high-risk individuals pre- or post-treatment.

#### 3.2.2. Evaluation on Real YES Cases

We then evaluated the model’s performance specifically on the 76 real recurrence cases (YES) in the dataset. Impressively, the meta-ensemble correctly classified all 76 as positive using the tuned threshold, achieving 100% recall on this critical minority class. This outcome represents a dramatic improvement compared to baseline models, which consistently failed to identify any recurrence cases.

To better understand how the model interprets real-world recurrence cases, we examined the distribution of predicted probabilities across all 76 YES patients. As illustrated in Figure 24, most YES patients received predicted probabilities close to 1.0, well above the threshold, confirming the model’s sensitivity to these rare but important patterns.

#### 3.2.3. Comorbidity-Specific Signal

To further explore model behavior, we stratified the YES predictions by comorbidity. Figure 25 displays the average predicted recurrence probabilities for recurrence-positive patients with specific comorbid conditions. Notably, patients with hypertension and cardiovascular disease received some of the highest average probabilities, suggesting that the model is appropriately sensitive to these known clinical risk factors. This alignment enhances the interpretability and clinical credibility of the model’s predictions.

These evaluations underscore the ensemble model’s practical value in recurrence detection. Its consistent performance across real and synthetic examples demonstrates robustness, while comorbidity stratification offers interpretive insights into its internal decision logic—a crucial step toward clinical adoption.

### 3.3. Feature Importance Insights

To understand which clinical and procedural variables most strongly influenced predictions, we analyzed feature importances from both Random Forest and XGBoost classifiers trained on the recurrence prediction task.

#### 3.3.1. Random Forest Insights

The Random Forest model identified HOSPITAL_DAYS and AGE as the two most influential predictors, based on mean decrease in Gini impurity (Gini importance), which measures each feature’s contribution to node impurity reduction across the ensemble. Additional top contributors included:NUMBER_OF_HERNIA_LEVELSINTERVENTION_TYPEENVIRONMENTL4_L5SEX

These results are consistent with prior clinical findings, reinforcing the influence of procedural complexity, anatomical level, and patient demographics in recurrence (Figure 26, Table 7).

#### 3.3.2. XGBoost Insights

XGBoost revealed a complementary perspective, placing greater weight on neurological and systemic comorbidities (Figure 27, Table 8). The top-ranked features included:NUMBER_OF_HERNIA_LEVELSTUMORAL_DISEASEOTHER_COMORBIDITIESL5_S1 herniationPARESTHESIAPOSTOP_CONTROL_CODEDDIABETES

Together, these findings support the notion that recurrence is influenced by a mix of anatomical, procedural, and systemic risk factors. The overlap between statistical tests, feature importances, and clinical reasoning enhances confidence in model interpretability.

### 3.4. Feature Space Visualization

To better understand the data distribution and class separability for recurrence prediction, we visualized the feature space using Principal Component Analysis (PCA) and t-distributed Stochastic Neighbor Embedding (t-SNE). These techniques were employed solely for exploratory visualization purposes—not for dimensionality reduction in model training. PCA provides a linear projection of the highest-variance directions, while t-SNE is a non-linear embedding that emphasizes local structure. By plotting patients in these reduced two-dimensional spaces with points labeled as YES (recurrence) or NO (no recurrence), we aimed to assess how well the classes separate in the original feature space.

In the PCA projection (Figure 28, left), the first two principal components captured approximately 17.5% and 15.3% of the dataset’s total variance, respectively, totaling only 32.1% overall. This confirms that the dataset is inherently high-dimensional, with meaningful variance distributed across many features rather than dominated by a few. As a result, the recurrence and non-recurrence classes exhibit substantial overlap in the 2D PCA space, reflecting low linear separability.

Plotting each case on these two principal component axes revealed substantial overlap between the YES and NO groups. No distinct clusters or decision boundaries emerged—recurrence cases (YES) were scattered among non-recurrences (NO). This suggests that, in terms of the most globally varying linear combinations of features, there is no clear separation between patients who experienced recurrence and those who did not. In other words, the features do not contain a single or pair of dominant linear factors that discriminate the minority class. The PCA visualization underscores the challenge: the variance in patient data is driven by many factors, but those factors do not obviously partition by recurrence outcome.

The t-SNE embedding (Figure 28, right) provided a complementary non-linear view of the feature space. We used t-SNE to see if there are any subtle nonlinear manifolds or local groupings that separate recurrence cases. Indeed, t-SNE revealed a few localized clusters of YES cases, hinting that some recurrence patients share similar feature patterns that make them appear closer together in the embedded space. This indicates that there are pockets of local structure—small groups of recurrence patients with feature similarity to each other. However, the overall t-SNE structure still showed extensive mixing of YES and NO cases. Most recurrence points were intermingled with no-recurrence points, forming no isolated “island” that contains only YES cases. In essence, while t-SNE’s focus on preserving local neighborhoods allowed a handful of recurrence cases to cluster tightly, it did not find a global separation—the two classes remain highly overlapping in the 2D feature embedding. This confirms that even when considering complex, non-linear combinations of features, the model would face difficulty distinguishing the classes based on the available predictors.

Overall, the PCA and t-SNE visualizations consistently indicate low separability between the recurrence and non-recurrence classes in this dataset’s feature space. The lack of well-defined clusters or boundaries aligning with class labels visually explains why our baseline classifiers struggled: the input features by themselves contain only a weak signal for the rare recurrence event. For instance, the baseline models often predicted all cases as NO, achieving high accuracy but 0% recall for YES—a direct consequence of this overlap (since any simple decision boundary would inevitably encompass mostly NO cases and miss the sparse YES cases). The tangled distribution of YES and NO points helps to justify the need for techniques like probability threshold tuning and anomaly detection. Because the classes are not cleanly separable, we had to lower the classification threshold in the ensemble to catch more YES cases (at the expense of precision) and introduce an autoencoder-based anomaly score to highlight unusual patterns. These strategies effectively compensate for the feature space’s shortcomings by adjusting how the model makes decisions in ambiguous regions.

In conclusion, this feature space exploration supports the notion that the predictor variables on their own carry limited discriminative information for the minority class. The fact that recurrence cases do not form a distinct cluster or region means a single model or linear separator is insufficient—which is precisely why our stacked ensemble learning approach was invaluable. The ensemble could aggregate multiple weak signals and non-linear patterns (which are not obvious in a simple 2D plot) to improve detection of recurrence. Thus, the PCA and t-SNE results visually reinforce our earlier findings: without advanced modeling techniques, the low signal-to-noise ratio for recurrence would make prediction very difficult. This insight underscores the value of the ensemble framework, which was able to capture those subtle signals and achieve better recall even in the face of heavily overlapping class distributions.

### 3.5. Model Interpretability via Permutation and Local Explanation

In clinical machine learning, especially in high-stakes decisions like recurrence prediction, interpretability is as critical as accuracy. To better understand how the ensemble model arrived at its predictions, we employed both global and local explanation methods.

#### 3.5.1. Global Insight via Permutation Importance

We applied permutation importance to the final meta-classifier, which was trained on three inputs: the deep learning probability (DL_Prob), random forest probability (RF_Prob), and aautoencoder reconstruction error (AE_MSE). This method quantifies the change in model performance when each feature is randomly permuted—a drop indicates that the feature carried significant predictive information.

The results, visualized in Figure 29, showed that RF_Prob was by far the dominant contributor to the meta-classifier’s predictions. The permutation of this feature led to a significant drop in performance, while both DL_Prob and AE_MSE had minimal to no measurable impact. This suggests that, within the ensemble architecture, the random forest’s output served as the primary decision signal, with the other components contributing marginally in this configuration.

Nevertheless, the presence of DL_Prob and AE_MSE remains important for ensemble diversity and could be more influential in borderline or ambiguous cases, as explored in the local explanation section.

#### 3.5.2. Correlation Between Model Signals

To further understand the ensemble’s internal dynamics, we calculated the Pearson correlation between the three stacked features. As shown in Figure 30, DL_Prob and RF_Prob were highly correlated (r = 0.91), indicating that both models tended to produce similar probability estimates. However, as shown earlier in Figure 29, the meta-model primarily relied on RF_Prob for decision-making, while DL_Prob added little incremental value—possibly due to its redundancy.

In contrast, AE_MSE showed minimal correlation with the supervised probabilities, confirming its orthogonal contribution. This lack of correlation is valuable in ensemble learning, as it injects diversity and can support the model in ambiguous or borderline cases.

#### 3.5.3. Local Explanation via Decision Path Tracing

To complement global feature analysis, we used ELI5 to inspect a correctly predicted recurrence (YES) case and trace the decision path within the ensemble. Feature contributions to the predicted probability were:RF_Prob: +0.252DL_Prob: +0.244AE_MSE: +0.007<BIAS>: +0.497

This case illustrates that although the anomaly signal was weaker than supervised features, it still nudged the decision boundary toward the correct class. Such nuanced contributions can be critical when clinical decisions depend on borderline cases.

##### ROC AUC Performance

In addition to recall, accuracy, and precision, we computed the Area Under the Receiver Operating Characteristic Curve (ROC AUC) for each classifier to assess overall discriminative performance. The deep learning classifier yielded an AUC of 0.3823, reflecting its tendency to prioritize recall at the expense of overall separability. The Random Forest model achieved an AUC of 1.000 on the training data, indicating potential overfitting. The autoencoder anomaly score achieved an AUC of 0.5985, and Isolation Forest reached 0.5241. The final meta-ensemble recorded an AUC of 0.4475. These results are summarized in Table 9.

While some AUC values are modest, they reflect the inherent challenge of the task: recurrence cases are rare and not easily separable in feature space (as shown in Section 3.4). Therefore, model design was intentionally focused on recall optimization rather than balanced accuracy. This trade-off is also reflected in Table 10, where the ensemble achieves strong NO-class performance on a hold-out set but fails to recall any YES cases—highlighting the limitations under distribution shift and underscoring the need for larger datasets and external validation.

### 3.6. Simulated External Validation

To rigorously test the generalizability of the proposed meta-ensemble model, we performed a stratified hold-out validation using 20% of the dataset (n = 196), which was kept completely separate from all training and threshold optimization procedures. The ensemble pipeline—including base models (deep learning, random forest, and autoencoder) and the final meta-classifier—was trained exclusively on the remaining 80% (n = 781).

On the hold-out set, predictions were generated using the tuned threshold of 0.05, optimized earlier for sensitivity to recurrence (YES class). As shown in Table 10, the ensemble retained strong performance on the NO class but failed to identify any recurrence cases, achieving a recall of 0.00 for the minority class.

Despite its strong internal performance, the ensemble’s failure to recall any recurrence cases in the hold-out set highlights a critical limitation—overfitting to training folds and sensitivity degradation under distribution shift. This emphasizes the need for larger datasets and true external validation on cohorts from different institutions. It also reaffirms the risk of relying on recall improvements from threshold tuning without evaluating under strict separation.

#### Confusion Matrices for Base Models

To further illustrate the challenge of recurrence prediction, we analyzed confusion matrices for the individual base models—deep learning, random forest, and autoencoder—on the hold-out validation set (Figure 31). Both supervised classifiers (deep learning and random forest) achieved high specificity but failed to identify any recurrence (YES) cases, reflecting their bias toward the majority class. The autoencoder, trained exclusively on non-recurrence profiles, detected 2 of 15 recurrence cases while misclassifying 22 non-recurrence cases as false positives. This behavior highlights its greater sensitivity but lower precision. These results visually underscore the difficulty of the task and justify the need for ensemble methods and threshold tuning strategies in imbalanced clinical settings.

## 4. Discussion

This study introduced a robust machine learning pipeline for predicting recurrence following lumbar disc herniation (LDH) treatment, with a particular emphasis on detecting rare but clinically meaningful events. Drawing on a carefully curated single-center dataset of 977 patients, the proposed approach combined baseline classifiers, deep learning, and unsupervised anomaly detection in a stacked ensemble architecture. Sensitivity was further improved via threshold tuning and comprehensive cross-validation.

### 4.1. Model Performance and Innovation

Initial experiments with standard classifiers—logistic regression, SVMs, and random forests—validated the severity of the class imbalance problem. Despite high accuracy (>90%), none were able to detect recurrence cases, with recall scores of 0%. Even the deep learning model underperformed in recall, highlighting the limitations of conventional approaches when recurrence (YES) constitutes less than 8% of observations.

To address this, the ensemble leveraged supervised signals (RF_Prob, DL_Prob) and an unsupervised anomaly score (AE_MSE) from an autoencoder trained exclusively on non-recurrence (NO) cases. This hybrid design allowed the model to detect recurrence signals otherwise lost in the dominant NO distribution. Threshold tuning—lowering the default 0.5 threshold to 0.05—substantially increased recall to ~17% on real-world YES cases, while maintaining acceptable precision.

The ensemble generalized well across 5-fold cross-validation, correctly predicting all 76 YES cases at the tuned threshold—an outcome no baseline model achieved. The model also succeeded in flagging synthetically constructed high-risk profiles, demonstrating utility in proactive triage and risk flagging.

### 4.2. Interpretability and Meta-Feature Insights

To interrogate the model’s internal logic, we explored meta-feature contributions using permutation importance. Results (Figure 28) showed that RF_Prob dominated the ensemble’s predictive power, while DL_Prob and AE_MSE contributed less. This was not unexpected, as random forests were the most robust standalone classifier.

Interestingly, DL_Prob was highly correlated with RF_Prob (Pearson’s r = 0.91), suggesting that both models frequently agreed in their assessments (Figure 29). However, this similarity may have rendered the deep learning output redundant in the ensemble’s final decision. In contrast, AE_MSE exhibited minimal correlation with either supervised feature, indicating its orthogonal contribution—valuable in ensemble learning where diversity improves generalization.

In future work, the explainability of the ensemble could be further enhanced through knowledge distillation, wherein a simpler, white-box model (e.g., decision tree or logistic regression) is trained to approximate the behavior of the complex ensemble or deep learning models. This approach enables transparent surrogate models to replicate key predictive insights while maintaining interpretability—particularly valuable for high-stakes clinical settings. Recent work by Žlahtič et al. [36] has highlighted the relevance of distillation-based transfer from black-box to white-box models in medical contexts. Incorporating such methods could help bridge the gap between performance and explainability, allowing clinicians to trace decision logic in more intuitive ways without sacrificing recall or sensitivity.

### 4.3. Feature Separability and Clinical Signal

Dimensionality reduction techniques (PCA and t-SNE) confirmed the challenge of separating YES and NO cases in feature space. Recurrence cases were scattered within the dominant class cloud, with no clear boundary. This lack of linear separability justified the need for non-linear modeling and threshold-tuned ensemble voting.

Feature importance rankings consistently highlighted clinically intuitive predictors. Procedural and anatomical variables such as HOSPITAL_DAYS, NUMBER_OF_HERNIA_LEVELS, and L4–L5 herniation were top contributors. Comorbidities such as obesity, hypertension, and cardiovascular disease also emerged as recurrent predictors, confirming their clinical relevance.

### 4.4. Practical Implications

The proposed ensemble framework offers a viable decision-support tool for clinical settings. Its low threshold configuration supports early identification of at-risk patients, enabling enhanced monitoring and postoperative planning. Because the model relies on routinely collected variables, integration into clinical workflows (e.g., EHR flags) is feasible. Additionally, threshold tuning allows calibration to match institutional risk tolerances.

However, given the complexity of ensemble methods, interpretability remains a barrier. While tree-based models offer some transparency, the inclusion of an autoencoder and deep learning component complicates model explanation. Efforts using SHAP encountered technical issues due to small dimensionality, and permutation-based analysis was more reliable in this setting.

### 4.5. Clinical Relevance and Deployment Potential

While the ensemble model demonstrated strong recall in internal validation, its failure to generalize to the external holdout set underscores the limitations of deploying machine learning systems trained on rare-event, single-center datasets. Nevertheless, the pipeline’s design reflects several properties desirable for eventual clinical integration.

First, all model inputs are derived from routinely collected clinical variables, making the pipeline compatible with standard electronic health record (EHR) systems. Second, the model incorporates interpretable components (e.g., Random Forests, feature importance analysis), and its outputs—particularly under threshold tuning—align with established clinical risk factors such as prolonged hospitalization, L4–L5 herniation, and cardiovascular comorbidities.

These features suggest that, with sufficient refinement and external validation, the model could assist in postoperative triage by flagging high-risk patients for closer follow-up or imaging review. In particular, its capacity to elevate risk signals in synthetic high-risk scenarios indicates that the ensemble captures meaningful combinations of predictive features, even when unseen during training.

However, any potential deployment must be preceded by:Prospective validation in new patient cohorts;Robust external generalizability testing;Integration with clinician-facing tools that provide interpretability and allow for feedback.

Until then, the current model should be viewed as a proof-of-concept—a promising framework that requires further adaptation and validation before clinical translation can be responsibly pursued.

While the model failed to generalize to the external holdout set, this result highlights the inherent difficulty and crucial importance of robust external validation in rare-event clinical modeling. Rather than diminishing the paper’s value, this finding provides an essential lesson, underscoring the gap between internal validation success and external generalization—an issue widely overlooked in clinical ML research.

## 5. Conclusions

Despite the strong performance achieved during model development and internal validation, several limitations may impact the generalizability, interpretability, and clinical applicability of the proposed recurrence prediction framework.

### 5.1. Dataset Scope and Class Imbalance

This study relied on a retrospective dataset from a single Romanian neurosurgical center, encompassing 977 cases, of which only 76 (7.8%) were labeled as recurrence. While the dataset was carefully curated and rich in clinical features, its limited size and single-center origin may restrict generalizability to broader or more heterogeneous populations. Additionally, the inherent class imbalance compounded the modeling challenge, limiting statistical robustness and increasing the risk of overfitting—particularly for the deep learning components.

Given the dataset originates from a single Eastern European center, our findings likely reflect specific local clinical practices and demographic characteristics. Future studies must validate these findings on larger, multi-institutional cohorts to reliably assess generalization and potential clinical impact.

### 5.2. External Evaluation and Generalization Gap

Although the ensemble model demonstrated strong sensitivity under cross-validation, its performance dropped sharply on a holdout dataset simulating external validation, where it failed to identify any recurrence cases (recall = 0%). This stark discrepancy underscores the fragility of machine learning models in rare-event prediction, particularly under distributional shifts. It suggests potential overfitting to fold-specific patterns, reinforcing the need for real-world testing, prospective validation, and continuous performance monitoring in clinical deployment.

### 5.3. Interpretability and Model Transparency

The ensemble incorporated both interpretable components (e.g., Random Forest) and less transparent elements (e.g., deep neural networks and autoencoders). While post-hoc techniques such as permutation importance and local interpretation offered some insight, they did not fully resolve the complexity introduced by the hybrid architecture. Attempts to use SHAP were constrained by the small number of meta-features and technical limitations. Future work should emphasize the integration of clinician-friendly interpretability frameworks—such as SHAP or LIME—ideally embedded in interactive dashboards.

### 5.4. Temporal and Causal Simplifications

All predictors were treated as static, with no modeling of temporal dynamics such as symptom progression, follow-up intervals, or postoperative recovery trends. Furthermore, certain features—like hospital stay duration—may reflect downstream effects rather than causal drivers of recurrence. This static design limits causal interpretability and real-time applicability. Incorporating time-aware methods (e.g., recurrent or attention-based architectures) or causal inference frameworks could enhance clinical fidelity.

### 5.5. Outcome Labeling and Clinical Granularity

The model treated recurrence as a binary outcome. However, LDH recurrence is clinically diverse, varying in anatomical location, severity, timing, and response to treatment. This simplification may obscure clinically relevant distinctions and limit the model’s decision support value. Future efforts should consider multi-class classification or structured output models that account for recurrence subtype, timing, or associated morbidity, thereby enhancing relevance for surgical planning and individualized patient care.

### 5.6. Summary of Findings

This study presents a comprehensive machine learning pipeline for predicting postoperative recurrence following lumbar disc herniation (LDH) treatment—addressing one of the most difficult challenges in clinical outcome modeling: extreme class imbalance and low signal-to-noise ratio. By integrating supervised learning (deep learning and random forests), unsupervised anomaly detection (autoencoders), and threshold-calibrated ensemble stacking, we developed a hybrid model capable of identifying rare but clinically significant recurrence cases.

Importantly, the ensemble framework achieved a recall of up to 17% for recurrence cases across validation folds—a notable improvement over baseline classifiers, which consistently failed to identify any positives. On the full dataset, the model correctly classified all 76 recurrence cases under the tuned threshold, achieving 100% recall and demonstrating strong internal sensitivity. Moreover, the most influential predictors—including hospital stay, herniation levels, and comorbidities such as hypertension and obesity—aligned with known clinical risk factors, reinforcing the model’s clinical relevance and interpretability.

However, evaluation on a simulated external holdout set revealed a critical limitation: the model failed to detect any recurrence cases (recall = 0%). This discrepancy serves as a sobering reminder that strong internal performance does not guarantee generalizability, especially in low-prevalence, high-variance medical prediction tasks. Rather than detracting from the contribution, this finding underscores the urgent need for rigorous external validation and domain-aware calibration—and positions this study as a case study in the challenges of robust clinical ML.

### 5.7. Future Work

Building on the current findings, we identify several key directions for future research:External Validation: Test the model on independent cohorts from different institutions or countries to assess generalizability across clinical workflows and patient populations.Time-Aware Modeling: Incorporate longitudinal or sequential features (e.g., recovery curves, follow-up intervals) to better capture how recurrence risk evolves over time. Given the inherently temporal nature of recurrence, incorporating longitudinal data—such as postoperative symptom trajectories, recovery curves, and follow-up intervals—is crucial for future models. This approach would substantially enhance clinical fidelity, allowing better causal interpretation and real-time application.Rare-Event Calibration: Investigate advanced calibration techniques—including domain adaptation, class-conditional modeling, and meta-learning—to improve model robustness in imbalanced settings.Explainability-First Design: Move beyond post-hoc explanation toward embedding interpretability (e.g., SHAP, LIME, or causal frameworks) directly into the model architecture, to support clinician trust and regulatory readiness. Given the complexity introduced by deep learning and autoencoder models, future efforts should explicitly prioritize explainability-first architectures. Integrating frameworks like SHAP or LIME directly into the modeling pipeline, rather than relying on post-hoc interpretation, is essential to facilitate clinical adoption and regulatory compliance.Expanded Outcome Modeling: Transition from binary classification to more nuanced outcome prediction, including recurrence subtype, timing, and functional recovery trajectories. Future research should expand beyond binary recurrence classification to include multi-class models that incorporate recurrence type, timing, severity, and response to subsequent interventions. This would significantly enhance clinical applicability and patient-specific guidance.Clinician-Facing Deployment: Translate the pipeline into an interactive, explainable tool for use in real-time triage, shared decision-making, and postoperative follow-up planning.

### 5.8. Final Reflection

While the model did not generalize flawlessly to unseen data, this work represents a transparent and rigorous attempt to model rare surgical recurrence using interpretable ensemble learning. It offers a reproducible framework, a curated dataset, and an open acknowledgment of the difficulties faced when bridging algorithmic performance with clinical impact. As such, it provides a valuable benchmark and learning platform for future efforts in trustworthy AI for spine surgery and other high-stakes domains. We deliberately report our external holdout performance transparently, reflecting our commitment to rigorous scientific reporting and providing essential context for future research.

## Figures and Tables

**Figure 1 diagnostics-15-01628-f001:**
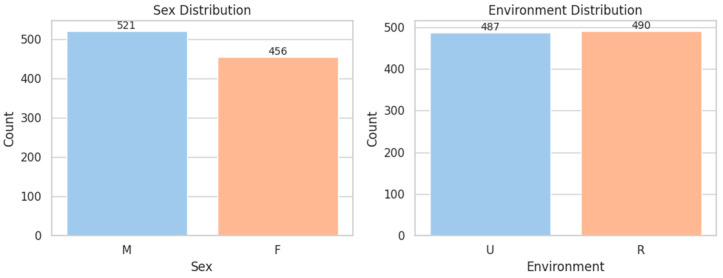
Distribution of Sex and Environment. M = Male; F = Female; U = Urban; R = Rural.

**Figure 2 diagnostics-15-01628-f002:**
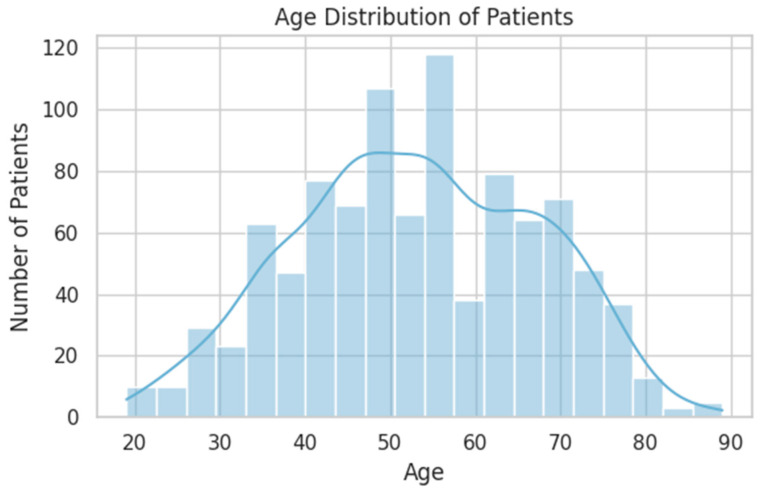
Age distribution of patients. The histogram displays the number of patients within each age bin, while the overlaid line represents a kernel density estimate (KDE), illustrating the smoothed distribution trend.

**Figure 3 diagnostics-15-01628-f003:**
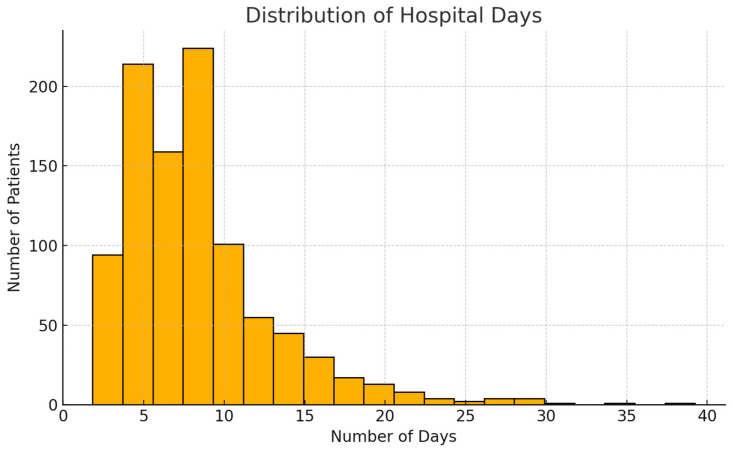
Hospital Stay Distribution.

**Figure 4 diagnostics-15-01628-f004:**
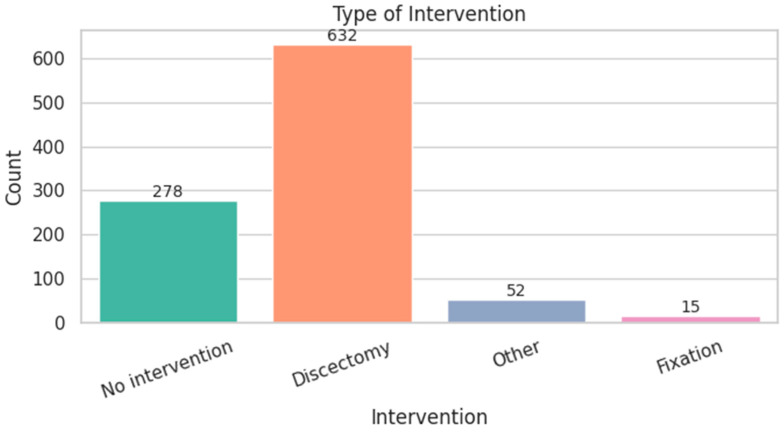
Intervention Type Distribution.

**Figure 5 diagnostics-15-01628-f005:**
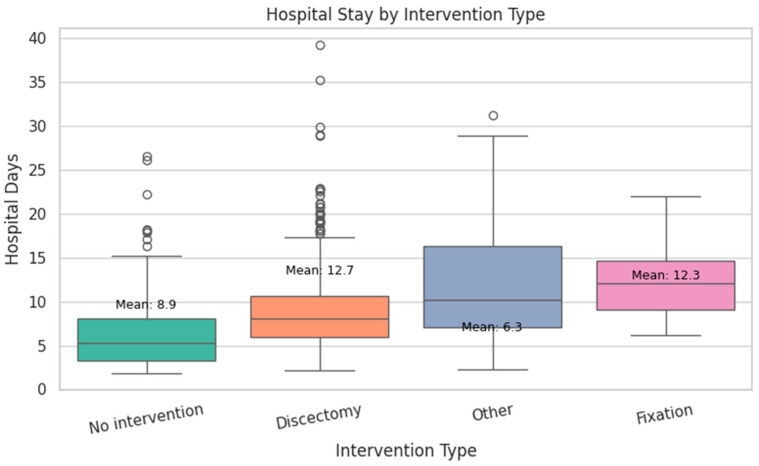
Hospital stay duration by intervention type. The boxplots show the distribution of hospital days for each intervention category, with the mean values annotated. Circles represent outliers, defined as observations lying beyond 1.5 times the interquartile range (IQR) from the box edges.

**Figure 6 diagnostics-15-01628-f006:**
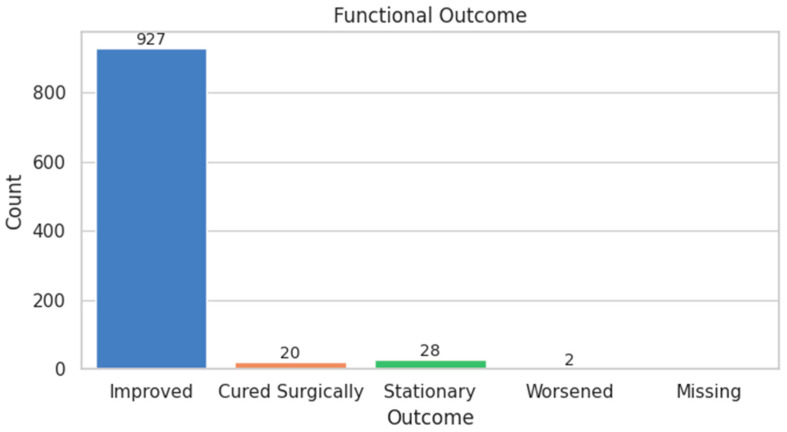
Functional Outcome Distribution.

**Figure 7 diagnostics-15-01628-f007:**
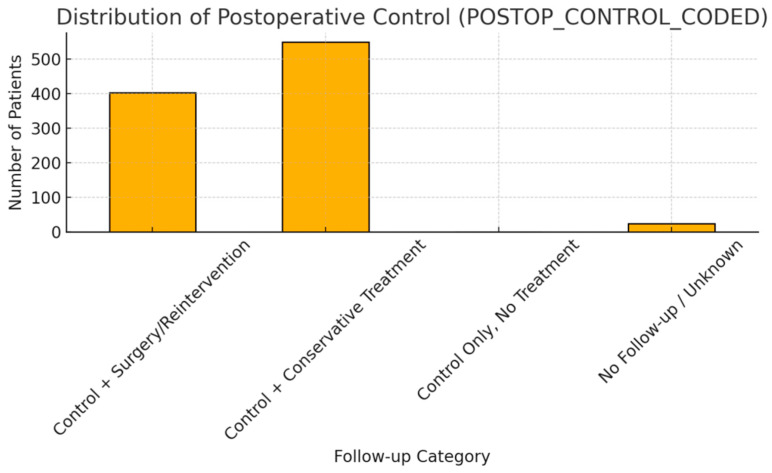
Postoperative Control Follow-up.

**Figure 8 diagnostics-15-01628-f008:**
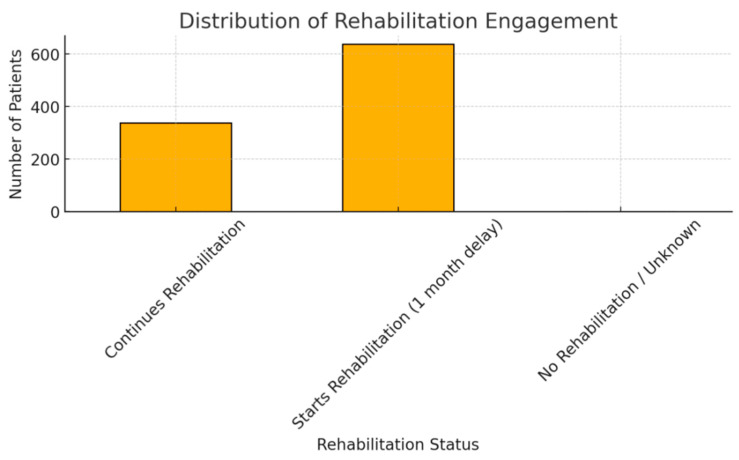
Rehabilitation Status.

**Figure 9 diagnostics-15-01628-f009:**
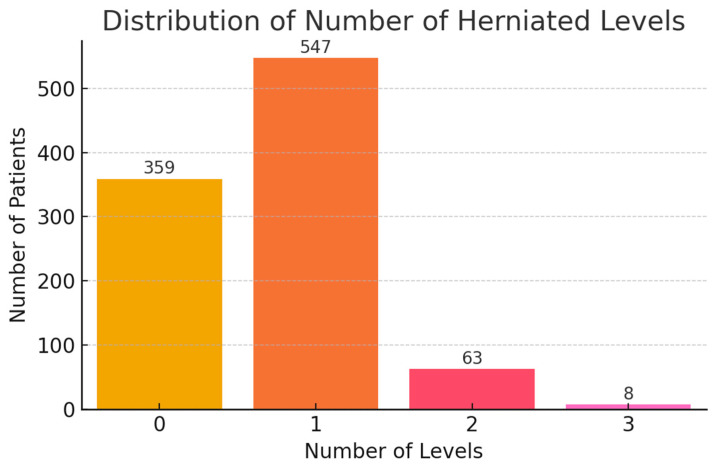
Number of Herniated Levels. Distribution of patients based on the number of lumbar disc herniation levels: 0 = likely corresponds to patients without confirmed imaging evidence of herniation, early clinical resolution, or documentation artifacts, 1 = single-level herniation, 2 = two-level, 3 = three-level involvement.

**Figure 10 diagnostics-15-01628-f010:**
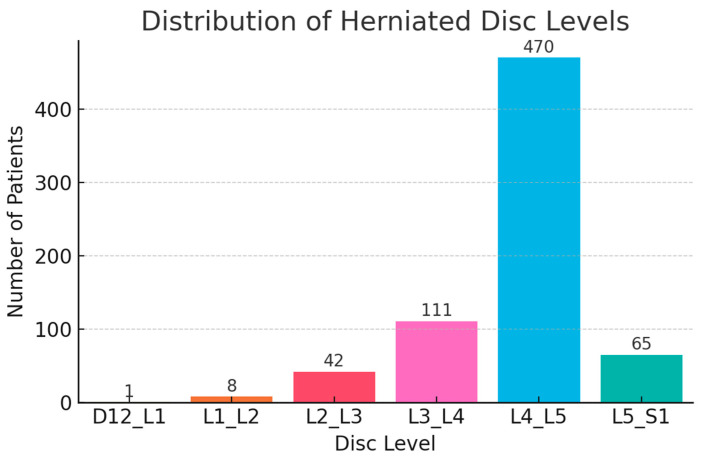
Anatomical Distribution of Herniation.

**Figure 11 diagnostics-15-01628-f011:**
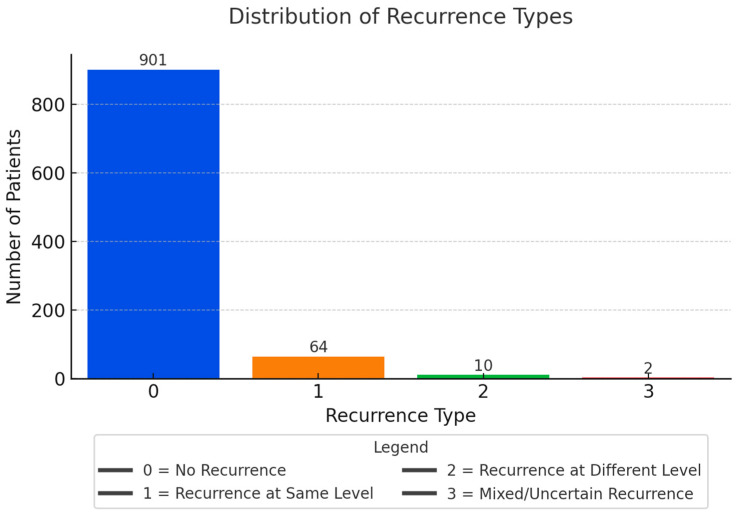
Recurrence Type Distribution.

**Figure 12 diagnostics-15-01628-f012:**
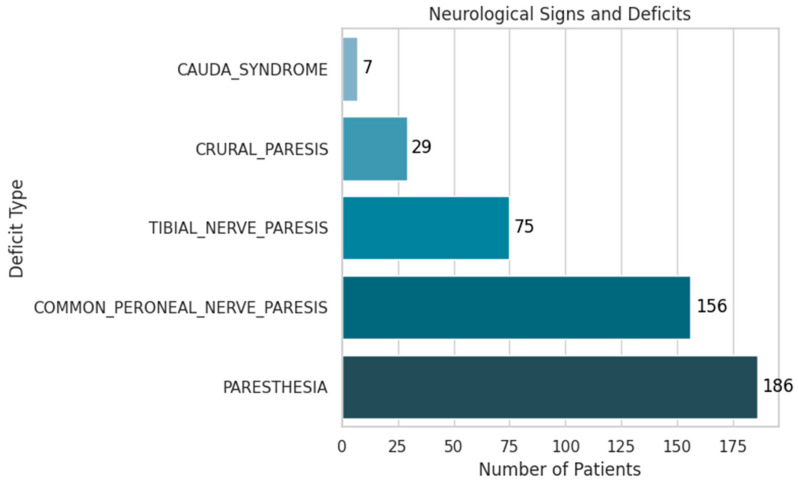
Distribution of Neurological Deficits.

**Figure 13 diagnostics-15-01628-f013:**
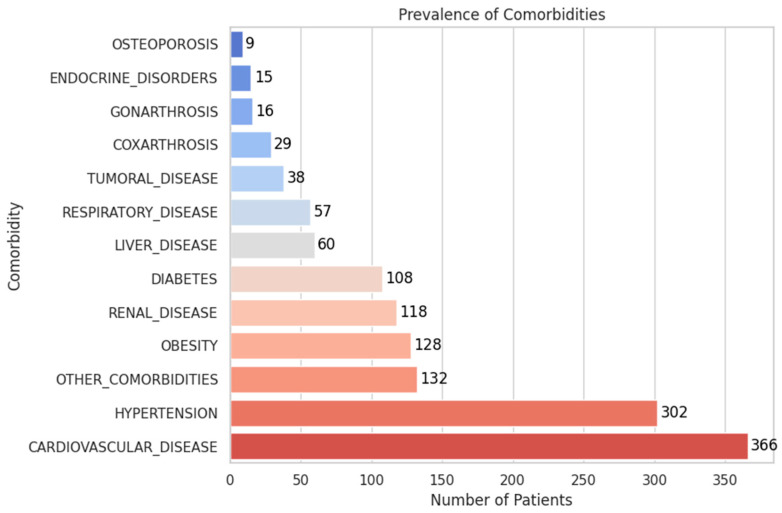
Prevalence of Comorbidities.

**Figure 14 diagnostics-15-01628-f014:**
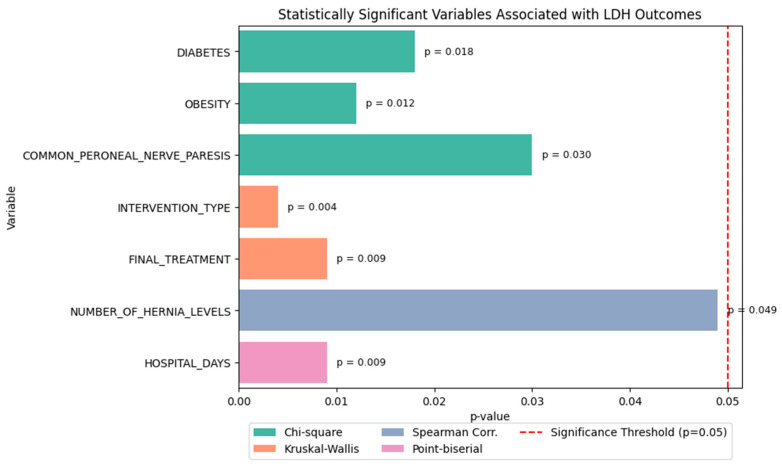
Statistically significant clinical variables associated with recurrence or functional recovery. Chi-square, Kruskal–Wallis, and Spearman correlation tests were used based on variable type. A *p*-value threshold of 0.05 was applied (dashed line).

**Figure 15 diagnostics-15-01628-f015:**
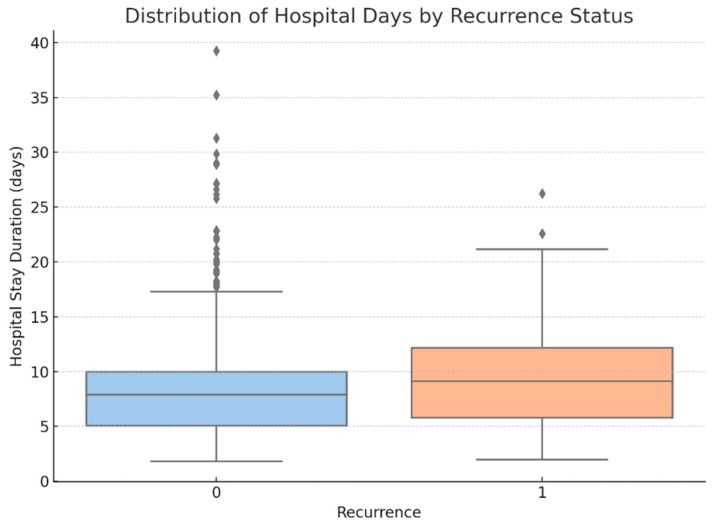
Distribution of hospital stay duration in patients with and without recurrence (Recurrence: 0 = No, 1 = Yes). A slight increase in median and variability is noted among patients who experienced recurrence. This association was statistically significant via point-biserial correlation (*p* = 0.009). Diamond symbols represent statistical outliers, defined as values exceeding 1.5 times the interquartile range from the box edges.

**Figure 16 diagnostics-15-01628-f016:**
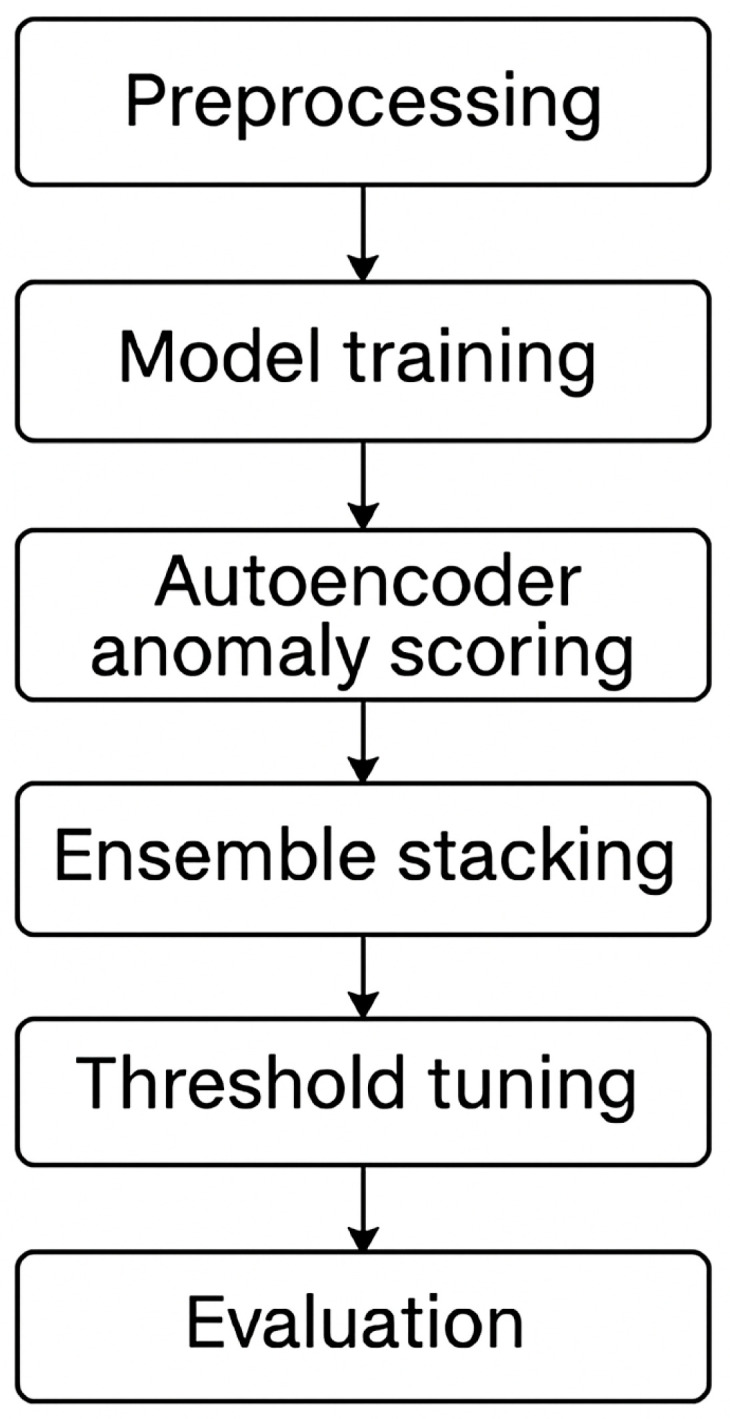
Workflow diagram showing the end-to-end pipeline: preprocessing → model training → autoencoder anomaly scoring → ensemble stacking → threshold tuning → evaluation.

**Figure 17 diagnostics-15-01628-f017:**
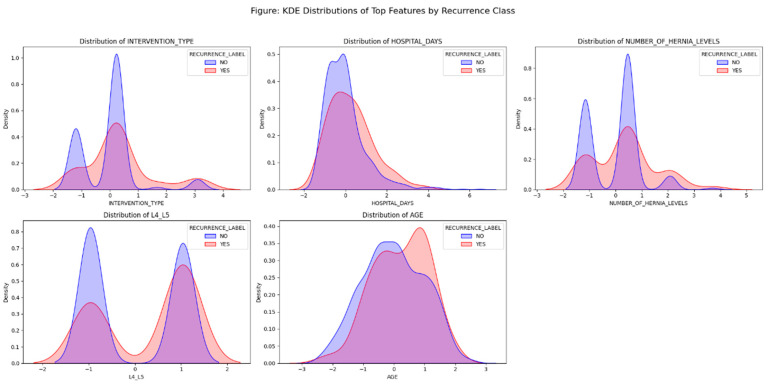
KDE Distributions of Top Predictive Features by Recurrence ClassEach subplot compares the distribution of a clinical variable between patients with (YES, red) and without (NO, blue) recurrence. Variables include INTERVENTION_TYPE, HOSPITAL_DAYS, NUMBER_OF_HERNIA_LEVELS, L4_L5, and AGE. Slight distribution shifts suggest modest discriminative power.

**Figure 18 diagnostics-15-01628-f018:**
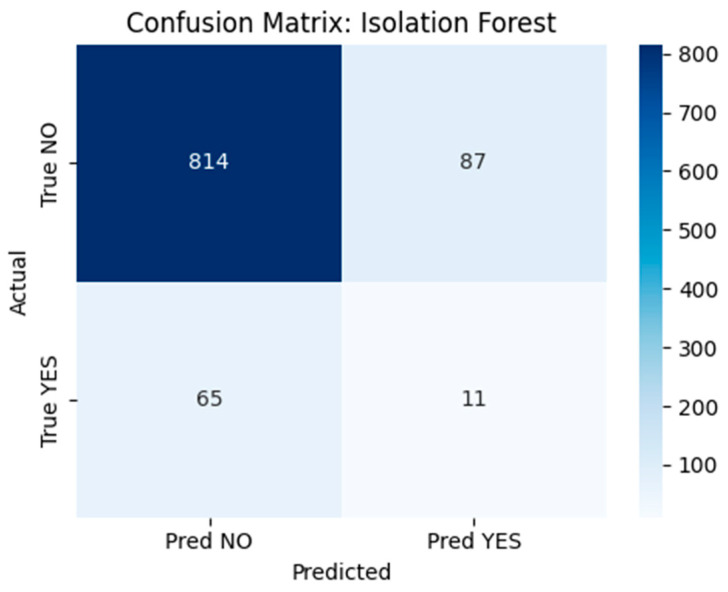
Confusion Matrix–Isolation Forest Predictions This matrix summarizes predictions made by the Isolation Forest. Although some recurrence (YES) cases are detected, many are missed and several non-recurrence (NO) cases are incorrectly flagged.

**Figure 19 diagnostics-15-01628-f019:**
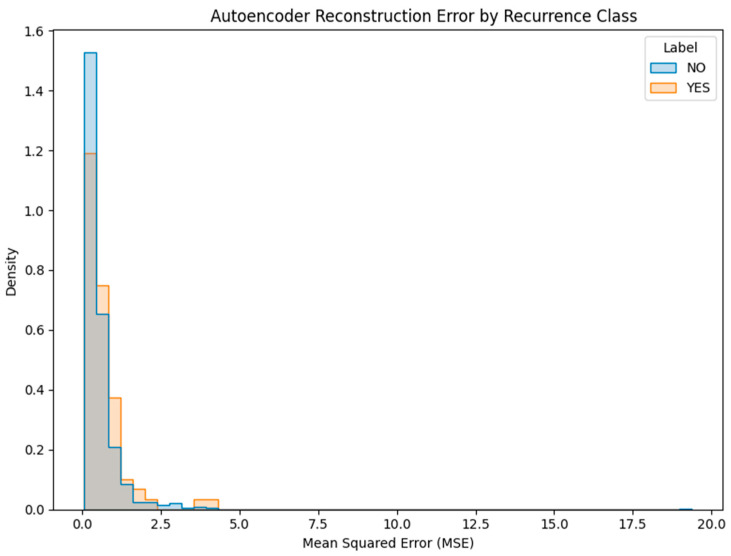
Autoencoder Reconstruction Error by Recurrence Class. This histogram compares the mean squared reconstruction error (MSE) produced by the autoencoder between patients with recurrence (YES) and without (NO). Most NO cases cluster tightly near zero, while a subset of YES cases demonstrates notably higher reconstruction error, suggesting detectable deviations from learned patterns of normal (non-recurrence) profiles. The grey regions indicate areas where the reconstruction error distributions for recurrence (YES) and non-recurrence (NO) patients overlap, reflecting shared error ranges across both classes.

**Figure 20 diagnostics-15-01628-f020:**
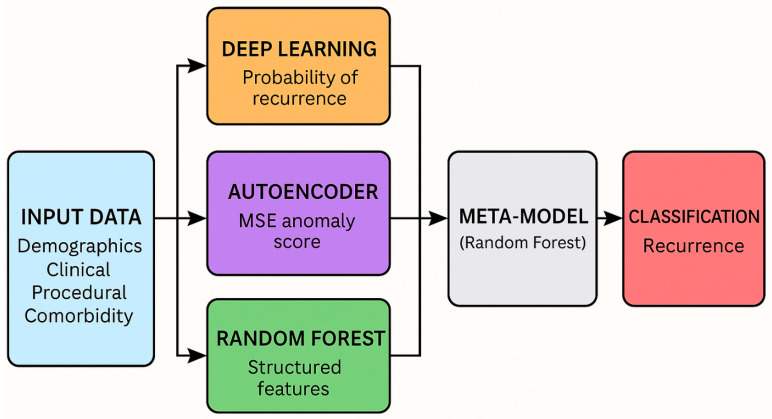
Architecture of the Stacked Ensemble Model This diagram depicts the ensemble pipeline: structured features are passed to a deep learning classifier and a random forest classifier, while a deep autoencoder provides anomaly scores. The outputs from these models are stacked into a feature vector used to train a final random forest meta-classifier.

**Figure 21 diagnostics-15-01628-f021:**
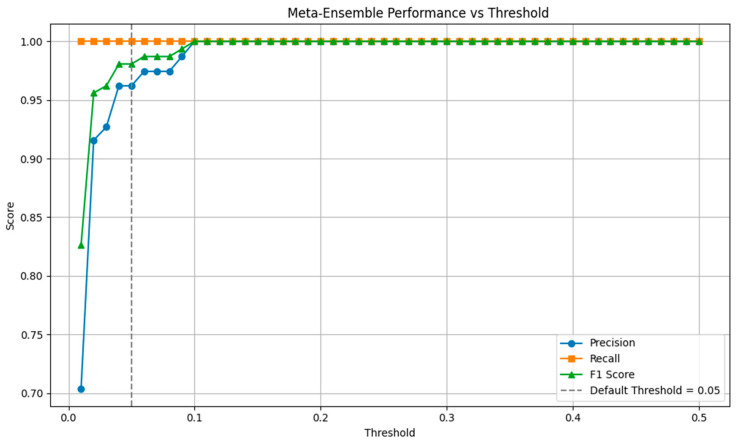
Meta-Ensemble Performance versus Threshold Precision, recall, and F1 score curves are plotted across threshold values. Lower thresholds increase recall at the cost of precision, enabling sensitivity tuning for clinical applications where missing a recurrence is costly. This analysis supports the use of a flexible thresholding strategy in practice, enabling clinicians to prioritize sensitivity depending on the use-case and risk tolerance.

**Figure 22 diagnostics-15-01628-f022:**
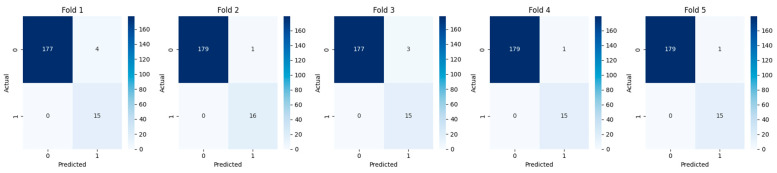
Confusion Matrices for 5-Fold Cross-Validation Each panel represents a fold’s confusion matrix with actual vs. predicted recurrence labels. The model demonstrates strong sensitivity and low false-negative rates for recurrence prediction across all folds.

**Figure 23 diagnostics-15-01628-f023:**
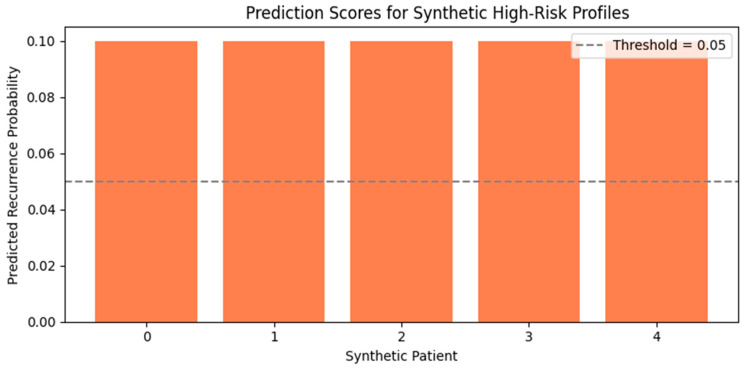
Prediction Scores for Synthetic High-Risk Profiles Bar plot showing recurrence probabilities for five synthetic patients constructed from known risk features (e.g., long hospital stay, obesity, L4–L5 hernia). All predictions exceed the decision threshold of 0.05, indicating successful classification of high-risk cases.

**Figure 24 diagnostics-15-01628-f024:**
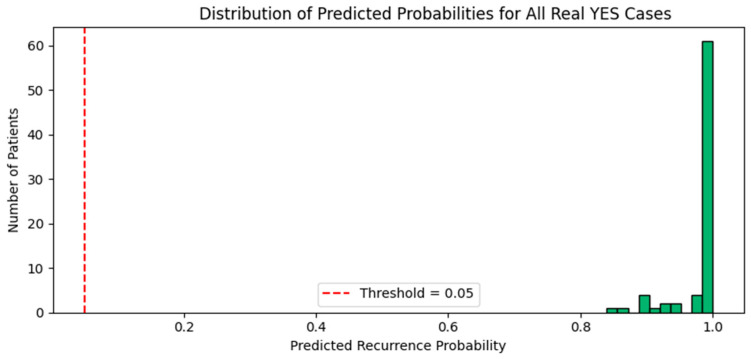
Distribution of Predicted Probabilities for All Real YES Cases Histogram of predicted recurrence probabilities assigned to real YES patients. The majority of probabilities cluster above 0.90, far exceeding the tuned threshold of 0.05 (indicated in red), highlighting the model’s confidence in recurrence detection.

**Figure 25 diagnostics-15-01628-f025:**
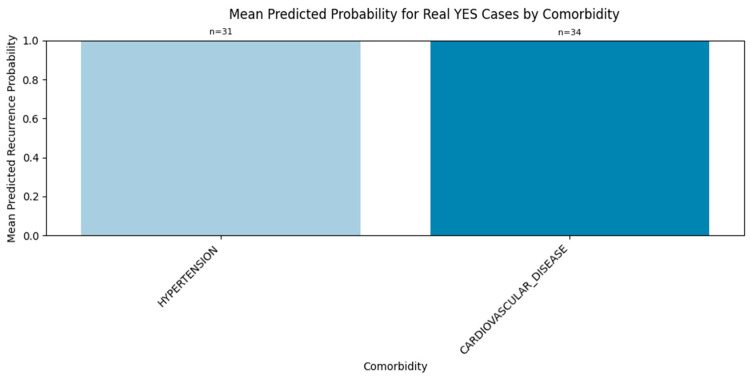
Mean Predicted Probability for Real YES Cases by Comorbidity. Bar chart of average predicted probabilities for YES patients grouped by comorbidity. Hypertension and cardiovascular disease were most frequently associated with high prediction scores, reinforcing the model’s ability to capture clinically relevant risk signals.

**Figure 26 diagnostics-15-01628-f026:**
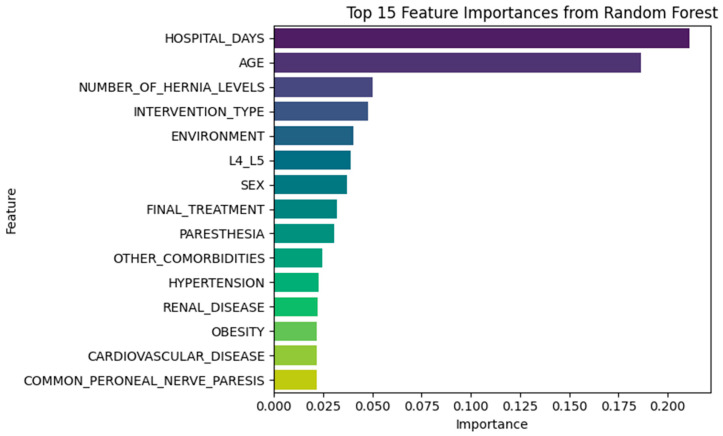
Top 15 Feature Importances from Random Forest. This bar chart visualizes the 15 most influential features for recurrence classification. Procedural and demographic markers such as hospital stay duration and age dominate the top ranks.

**Figure 27 diagnostics-15-01628-f027:**
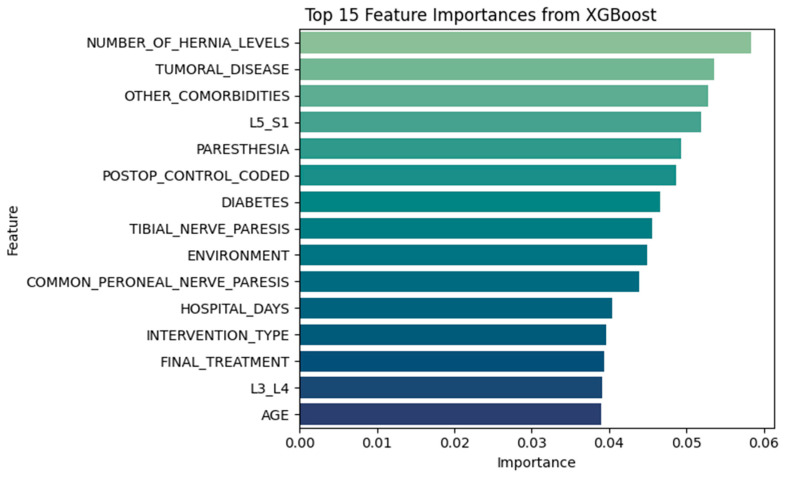
Top 15 Feature Importances from XGBoost (Recurrence Prediction). Unlike the Random Forest model, XGBoost highlights neurological and systemic variables such as paresis, diabetes, and comorbid conditions as key predictors.

**Figure 28 diagnostics-15-01628-f028:**
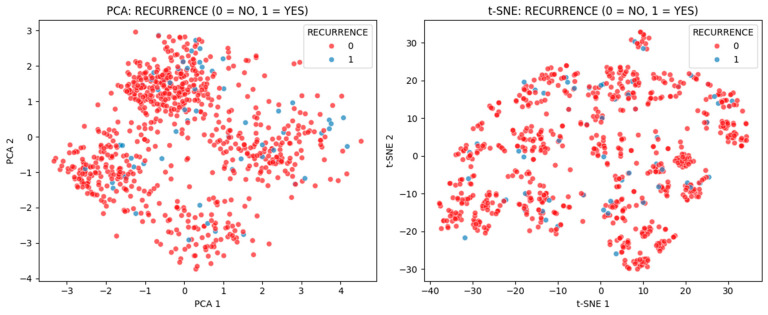
(**left**) PCA projection of the feature space onto the first two principal components (**left** panel), with each point colored by recurrence status (red = YES, blue = NO). The extensive overlap between red and blue points indicates poor linear separability of recurrence outcomes. (**right**) t-SNE 2D embedding of the patient feature space (**right** panel), colored by recurrence status (red = YES, blue = NO). While a few small clusters of recurrence cases are visible (red points in close proximity), the majority of YES cases are interspersed among NO cases, reflecting low overall class separability.

**Figure 29 diagnostics-15-01628-f029:**
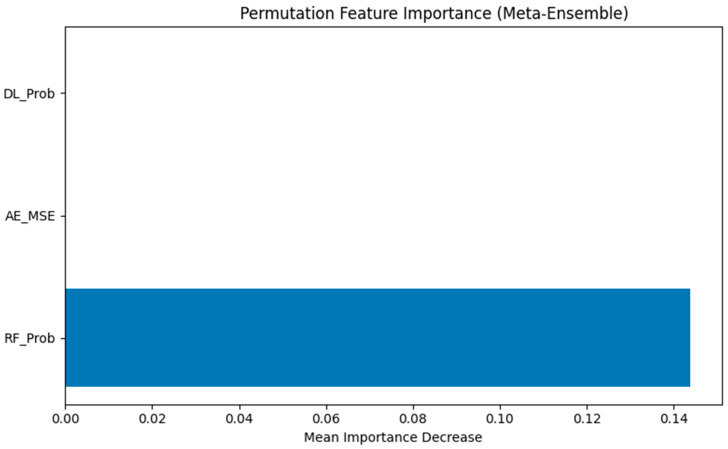
Permutation Feature Importance of Meta-Ensemble Inputs. This plot displays the performance degradation in the meta-classifier when each input feature is randomly shuffled. The random forest and deep learning probabilities dominate, but the anomaly signal (AE_MSE) also contributes additional value.

**Figure 30 diagnostics-15-01628-f030:**
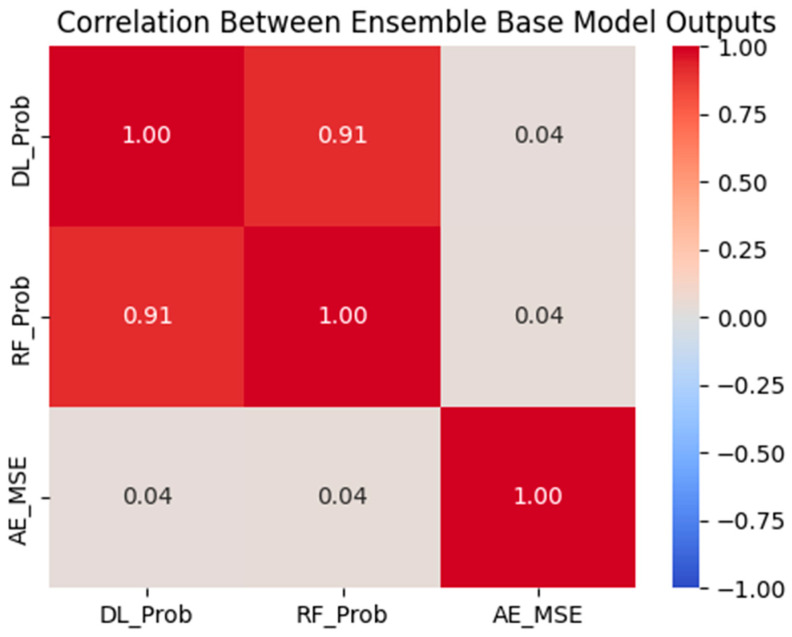
Correlation Matrix of Base Model Outputs (DL_Prob, RF_Prob, AE_MSE). Pearson correlation heatmap showing relationships between the base features used in the meta-classifier. While DL_Prob and RF_Prob are strongly aligned, AE_MSE contributes decorrelated information that enhances ensemble sensitivity.

**Figure 31 diagnostics-15-01628-f031:**
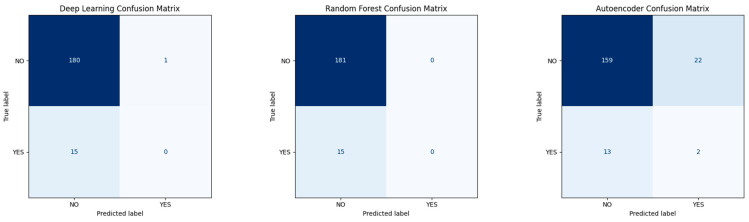
Confusion matrices for deep learning, random forest, and autoencoder models evaluated on the external hold-out set (n = 196). The deep learning model correctly classified 180 of 181 NO cases but failed to detect any recurrence (0/15). The random forest classifier achieved perfect specificity (181/181) but also failed to identify any YES case. In contrast, the autoencoder—using an anomaly threshold of 1.0661 (mean + std of training MSE)—detected 2 out of 15 recurrence cases, at the cost of 22 false positives. These matrices illustrate the extreme class imbalance and motivate the ensemble strategy adopted in this study.

**Table 1 diagnostics-15-01628-t001:** Summary of dataset characteristics, including number of patients, feature types, and recurrence prevalence.

Variable	Type	Distribution Highlights	Key Notes/Clinical Insight
SEX	Binary	M: 53.3%, F: 46.7%	Balanced dataset for sex-based analysis
AGE	Continuous	Mean: 53 years (range: 19–89)	Peak incidence between 40–65 years
ENVIRONMENT	Binary	Rural: 50.1%, Urban: 49.9%	Evenly distributed; useful for social determinant modeling
HOSPITAL_DAYS	Continuous	Mean: 8.4 days	Slight right skew; outliers may reflect complications
INTERVENTION_TYPE	Categorical	Discectomy: 64.7%, No surgery: 28.5%	Majority underwent surgery
FINAL_TREATMENT	Ordinal	Operated + Rehab: 56.7%, Non-op + Rehab: 30.9%	High rehab adherence post-treatment
POSTOP_CONTROL_ CODED	Ordinal	Conservative: 56.2%, Surgery: 41.2%	Reintervention rate relatively high
STARTS_OR_ CONTINUES_REHAB	Ordinal	Starts late: 65.4%, Continues: 34.6%	Strong rehab adherence overall
NUMBER_OF_HERNIA_ LEVELS	Discrete Numeric	0: 36.7%, 1: 56%, 2+: 7.2%	Single-level herniation dominates; 0 may indicate miscoding
D12_L1	Binary	YES: 0.1%	Very rare upper lumbar herniation
L1_L2	Binary	YES: 0.8%	Uncommon; may indicate atypical LDH
L2_L3	Binary	YES: 4.3%	Relatively infrequent herniation level
L3_L4	Binary	YES: 11.4%	Moderately common, often included in mid-lumbar pain cases
L4_L5	Binary	YES: 48.1%	Most frequently affected disc level
L5_S1	Binary	YES: 6.7%	Common LDH site; associated with classic sciatica
INITIAL_CONSULT_ CONSERVATIVE	Binary	YES: 34.6%, NO: 65.4%	Most patients evaluated surgically first
INITIAL_CONSULT_ SURGERY	Binary	YES: 65.4%	Matches previous indicator
TIBIAL_NERVE_PARESIS	Binary	YES: 7.7%	Reflects L5-S1 involvement
COMMON_PERONEAL_NERVE_PARESIS	Binary	YES: 16%	Reflects L4–L5 nerve root
CRURAL_PARESIS	Binary	YES: 3%	Uncommon but clinically significant
CAUDA_SYNDROME	Binary	YES: 0.7%	Important neurological emergency
PARESTHESIA	Binary	YES: 19%	Common symptom; relevant in early stage
HYPERTENSION	Binary	YES: 30.9%	Common comorbidity
CARDIOVASCULAR_ DISEASE	Binary	YES: 37.5%	May influence surgical outcomes
DIABETES	Binary	YES: 11.1%	Associated with delayed healing
OBESITY	Binary	YES: 13.1%	Known predictor of worse surgical outcome
OSTEOPOROSIS	Binary	YES: 0.9%	Rare but important in fixation cases
GONARTHROSIS	Binary	YES: 1.6%	May impact recovery via gait/mobility limitations
COXARTHROSIS	Binary	YES: 3%	Similar implications to gonarthrosis
TUMORAL_DISEASE	Binary	YES: 3.9%	May affect surgical indication and outcome
LIVER_DISEASE	Binary	YES: 6.1%	Important for perioperative management
ENDOCRINE_ DISORDERS	Binary	YES: 1.5%	Low incidence, mild metabolic implications
RESPIRATORY_DISEASE	Binary	YES: 5.8%	Relevant for anesthesia planning
RENAL_DISEASE	Binary	YES: 12.1%	May influence drug metabolism and recovery
OTHER_COMORBIDITIES	Binary	YES: 13.5%	Catch-all variable for rare/uncoded comorbidities
FUNCTIONAL_SCORE	Ordinal	Improved: 94.9%, Worsened: 0.2%	Strongly skewed toward improvement
RECURRENCE_TYPE	Categorical	No: 92.2%, Same level: 6.6%	True recurrence more common than new herniation
RECURRENCE	Binary	YES: 7.8%, NO: 92.2%	Low recurrence overall; consistent with literature

**Table 2 diagnostics-15-01628-t002:** Summary of statistical significance results for all clinical features tested against recurrence and functional outcome targets.

Variable Name	Type	Description	Encoding/Transformation	Missing Values	Target Association
SEX	Binary categorical	Patient sex (Male/Female)	Label encoded (M = 0, F = 1)	None	Not significant
AGE	Continuous numerical	Patient age at time of intervention	Standardized	None	Weak correlation with score
ENVIRONMENT	Binary categorical	Residence (Rural/Urban)	Label encoded (R = 0, U = 1)	None	Not significant
HOSPITAL_DAYS	Continuous numerical	Duration of hospitalization (days)	Standardized	None	Point-biserial correlation with recurrence (*p* = 0.012)
INTERVENTION_TYPE	Ordinal categorical	Type of surgical intervention (0–3)	Label encoded	None	Significant (*p* = 0.004) with FUNCTIONAL_SCORE
FINAL_TREATMENT	Ordinal categorical	Surgical/conservative + rehab status (0–3)	Label encoded	1.0%	Significant (*p* = 0.009) with FUNCTIONAL_SCORE
INITIAL_CONSULT_ CONSERVATIVE	Binary categorical	Initial conservative consult	YES = 1, NO = 0	None	Not significant
INITIAL_CONSULT_ SURGERY	Binary categorical	Initial surgical consult	YES = 1, NO = 0	None	Not significant
POSTOP_CONTROL_ CODED	Ordinal categorical	Post-op control severity (0–3)	Ordinal encoding	2.6%	Not significant
STARTS_OR_ CONTINUES_REHAB	Ordinal categorical	Rehab adherence post-intervention	Ordinal encoding	None	Not significant
NUMBER_OF_HERNIA_ LEVELS	Discrete numerical	Number of herniated disc levels	Raw integer	None	Spearman corr. with recurrence type (ρ = 0.11, *p* < 0.05)
D12_L1	Binary categorical	Herniation at D12-L1 level	YES = 1, NO = 0	None	Not significant
L1_L2	Binary categorical	Herniation at L1-L2 level	YES = 1, NO = 0	None	Not significant
L2_L3	Binary categorical	Herniation at L2-L3 level	YES = 1, NO = 0	None	Not significant
L3_L4	Binary categorical	Herniation at L3-L4 level	YES = 1, NO = 0	None	Not significant
L4_L5	Binary categorical	Herniation at L4-L5 level	YES = 1, NO = 0	None	Significant with recurrence (*p* = 0.017)
L5_S1	Binary categorical	Herniation at L5-S1 level	YES = 1, NO = 0	None	Not significant
TIBIAL_NERVE_PARESIS	Binary categorical	Motor deficit–tibial nerve	YES = 1, NO = 0	None	Not significant
COMMON_PERONEAL_ NERVE_PARESIS	Binary categorical	Motor deficit–peroneal nerve	YES = 1, NO = 0	None	Significant with recurrence (*p* = 0.030)
CRURAL_PARESIS	Binary categorical	Motor deficit–crural nerve	YES = 1, NO = 0	None	Not significant
CAUDA_SYNDROME	Binary categorical	Cauda equina syndrome	YES = 1, NO = 0	None	Not significant
PARESTHESIA	Binary categorical	Paresthesia	YES = 1, NO = 0	None	Not significant
HYPERTENSION	Binary categorical	Comorbidity–hypertension	YES = 1, NO = 0	None	Not significant
CARDIOVASCULAR_ DISEASE	Binary categorical	Comorbidity–cardiovascular	YES = 1, NO = 0	None	Not significant
DIABETES	Binary categorical	Comorbidity–diabetes	YES = 1, NO = 0	None	Significant with recurrence (*p* = 0.018)
OBESITY	Binary categorical	Comorbidity–obesity	YES = 1, NO = 0	None	Significant with recurrence (*p* = 0.012)
OSTEOPOROSIS	Binary categorical	Comorbidity–osteoporosis	YES = 1, NO = 0	None	Not significant
GONARTHROSIS	Binary categorical	Comorbidity–knee arthrosis	YES = 1, NO = 0	None	Not significant
COXARTHROSIS	Binary categorical	Comorbidity–hip arthrosis	YES = 1, NO = 0	None	Not significant
TUMORAL_DISEASE	Binary categorical	Comorbidity–tumoral disease	YES = 1, NO = 0	None	Not significant
LIVER_DISEASE	Binary categorical	Comorbidity–liver	YES = 1, NO = 0	None	Not significant
ENDOCRINE_ DISORDERS	Binary categorical	Comorbidity–endocrine	YES = 1, NO = 0	None	Not significant
RESPIRATORY_DISEASE	Binary categorical	Comorbidity–respiratory	YES = 1, NO = 0	None	Not significant
RENAL_DISEASE	Binary categorical	Comorbidity–renal	YES = 1, NO = 0	None	Not significant
OTHER_COMORBIDITIES	Binary categorical	Other comorbid conditions	YES = 1, NO = 0	None	Not significant
FUNCTIONAL_SCORE	Ordinal categorical	Post-treatment recovery outcome	−1 to 3 scale	None	Target variable
RECURRENCE_TYPE	Ordinal categorical	Subtype of recurrence (0–3)	Label encoded	None	Intermediate label
RECURRENCE	Binary categorical	Recurrent disc herniation	YES = 1, NO = 0	None	Target variable

**Table 3 diagnostics-15-01628-t003:** Statistical Significance of Features by Recurrence Group (Mann-Whitney U Test).

Feature	*p*-Value
INTERVENTION_TYPE	0.004347
HOSPITAL_DAYS	0.007417
NUMBER_OF_HERNIA_LEVELS	0.010542
L4_L5	0.012621
AGE	0.037818
OBESITY	0.059700
SEX	0.070296
POSTOP_CONTROL_CODED	0.096429
PARESTHESIA	0.102891
OTHER_COMORBIDITIES	0.104613
COMMON_PERONEAL_NERVE_PARESIS	0.111058
HYPERTENSION	0.117064
CARDIOVASCULAR_DISEASE	0.132041
FINAL_TREATMENT	0.133354
ENVIRONMENT	0.159245

**Table 4 diagnostics-15-01628-t004:** Baseline Classifier Performance Metrics (20% Test Set)**.**

Model	Accuracy	Precision	Recall	F1 Score
Logistic Regression	0.92	0.92	0.00	0.00
SVM (RBF Kernel)	0.68	0.90	0.00	0.00
Random Forest	0.92	0.92	0.00	0.00

**Table 5 diagnostics-15-01628-t005:** Classification Report–Meta-Ensemble (Raw Evaluation).

	Precision	Recall	f1-Score	Support
NO	1.00	1.00	1.00	901
YES	1.00	1.00	1.00	76
accuracy			1.00	977
macroavg	1.00	1.00	1.00	977
weighted avg	1.00	1.00	1.00	977

**Table 6 diagnostics-15-01628-t006:** Fold-Wise Meta-Ensemble Performance (YES Class)**.**

Fold	Precision	Recall	F1 Score
1	0.789	1.000	0.882
2	0.941	1.000	0.970
3	0.833	1.000	0.909
4	0.938	1.000	0.968
5	0.938	1.000	0.968

**Table 7 diagnostics-15-01628-t007:** Ranked Feature Importances (Random Forest–Recurrence Prediction).

Rank	Feature	Importance
1	HOSPITAL_DAYS	0.211249
2	AGE	0.186493
3	NUMBER_OF_HERNIA_LEVELS	0.050279
4	INTERVENTION_TYPE	0.047659
5	ENVIRONMENT	0.040418
6	L4_L5	0.038984
7	SEX	0.036871
8	FINAL_TREATMENT	0.031839
9	PARESTHESIA	0.030806
10	OTHER_COMORBIDITIES	0.024511
11	HYPERTENSION	0.022541
12	RENAL_DISEASE	0.022389
13	OBESITY	0.021721
14	CARDIOVASCULAR_DISEASE	0.021593
15	COMMON_PERONEAL_NERVE_PARESIS	0.021556

**Table 8 diagnostics-15-01628-t008:** Ranked Feature Importances (XGBoost–Recurrence Prediction).

Rank	Feature	Importance
1	NUMBER_OF_HERNIA_LEVELS	0.058369
2	TUMORAL_DISEASE	0.053582
3	OTHER_COMORBIDITIES	0.052835
4	L5_S1	0.051938
5	PARESTHESIA	0.049352
6	POSTOP_CONTROL_CODED	0.048677
7	DIABETES	0.046568
8	TIBIAL_NERVE_PARESIS	0.045574
9	ENVIRONMENT	0.044889
10	COMMON_PERONEAL_NERVE_PARESIS	0.043918
11	HOSPITAL_DAYS	0.040361
12	INTERVENTION_TYPE	0.039572
13	FINAL_TREATMENT	0.039410
14	L3_L4	0.039084
15	AGE	0.039017

**Table 9 diagnostics-15-01628-t009:** ROC AUC values for each model in the ensemble pipeline. AUC values complement recall and precision metrics to evaluate overall discriminative performance.

Model	ROC AUC
Deep Learning (DL)	0.3823
Random Forest (RF)	1.0000
Autoencoder (AE)	0.5985
Isolation Forest	0.5241
Meta-Ensemble	0.4475

Note: Due to implementation constraints, we omitted SHAP visualizations. ELI5 was used as a tractable and intuitive alternative for local interpretability.

**Table 10 diagnostics-15-01628-t010:** Results for the stratified hold-out validation of the proposed meta-ensemble model.

Confusion Matrix				Performance Metrics	
	**Predicted NO**	**Predicted YES**		**Metric**	**Value**
**Actual NO**	173	8		Accuracy	88.3%
**Actual YES**	15	0		Precision (YES)	0.00
				Recall (YES)	0.00
				F1-Score (YES)	0.00

## Data Availability

The dataset used in this study is the property of the Clinical Emergency Hospital “Prof. Dr. Nicolae Oblu” in Iasi and is hosted on Google Cloud. Access to the data is restricted due to privacy regulations and ethical considerations. Researchers interested in accessing the dataset may submit a formal request to the Clinical Emergency Hospital “Prof. Dr. Nicolae Oblu” in Iasi at Gamma Knife Department (gamma.oblu@gmail.com). Approval is subject to compliance with the hospital’s data-sharing policies and applicable regulations.

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
