# Peer review of "A Hybrid Ensemble Learning Framework for Predicting Lumbar Disc Herniation Recurrence: Integrating Supervised Models, Anomaly Detection, and Threshold Optimization"

_diagnostics, 2025, doi:10.3390/diagnostics15131628_

Round 1

Reviewer 1 Report

Comments and Suggestions for Authors

The Authors propose an elaborate strategy, based on the use of Machine Learning (ML) tools, to
predict the Lumbar Disc Herniation recurrence based on a dataset featuring clinical patients’ characteristics, such as age, herniation level, number of herniation levels, obesity, etc. The ML models reached high levels of recall, effectively addressing the problem of detecting rare recurrence cases. Despite the completeness of the results presented in the manuscript and the appropriateness of the contents for the scope of the journal, I recommend publishing the manuscript in this journal only after major revisions. Generally speaking, I suggest making the manuscript more concise to increase its readability. In particular:
1. I suggest merging subsection 4.6 with the Introduction, to emphasize the limits of prior works
and to introduce the innovative character of the study;
2. analogously, I suggest merging sections 5 and 6 into a unique Conclusions section;
3. It seems that Section 3, which presents the Results, also includes the description of the Machine Learning algorithms and pipeline in general. To increase the readability, I suggest organizing the description of the Machine Learning procedures and Data analysis in general within a subsection separated from the Results, merged with the Materials and Methods section.
4. Caption of Fig. 1: even if the meaning could appear obvious, I suggest anyway to indicate clearly the meaning of the letters M, F, U, and R.
5. Row 181: Which criterion is employed to establish whether a patient has improved or a stationary outcome? Please specify.
6. Caption of Fig. 9: I didn’t catch the meaning of 0 herniated levels. Please clarify.
7. Fig. 14: Could you please specify clearly within the text the reason behind the use of parametric
and non-parametric tests for different variables?
8. Fig. 15: The meaning of the values 0 and 1 of recurrence is explained in Table 2, which,
however, is reported on the following page. I suggest indicating the values of the variable as text
or indicating the meaning of the numeric values within the caption.
9. Row 402: Is there a particular reason behind the choice of 100 trees for the Random Forest
Classifier? Could you add details about the building of this model, e.g., eventual limits on the
single tree depth?
10. Rows 420-426: Again, I suggest adding further details about the properties of the forest, e.g., the number of trees , to increase the reproducibility of the methods.
11. Row 453: to be clearer, I suggest adding the extended name of the network, i.e., Multi-Layer
Perceptron (MLP), and to specify the model characteristics, i.e., number of layers, etc.
12. Please specify the criterion adopted to determine the Feature Importance reported in Fig. 26
(Gini index?).
13. Rows 631-633: Could the Authors report the Cumulative Explained Variance Ratio associated
with the first two principal components?
To conclude, due to the highly elaborate ML routine, I want to stress the need to add more details and to improve the general quality of the description of the Methods related to this aspect, to make the study more reproducible.

Author Response

Subject: Response to Reviewer 1 – Revised Manuscript Submission for Diagnostics, Manuscript ID: 3631107

Dear Reviewer,

On behalf of all co-authors, I would like to extend our sincere thanks for your thorough and thoughtful review of our manuscript, titled "A Hybrid Ensemble Learning Framework for Predicting Lumbar Disc Herniation Recurrence: Integrating Supervised Models, Anomaly Detection, and Threshold Optimization", submitted to Diagnostics (MDPI). Your critical feedback and insightful suggestions have greatly contributed to improving the quality, clarity, and reproducibility of our study.

We have carefully considered each of your comments and have made corresponding revisions throughout the manuscript. In what follows, we provide a detailed, point-by-point response outlining the changes implemented.

  1. Merge of Subsection 4.6 with the Introduction

We have integrated the full content of Subsection 4.6 into the Introduction, with the exception of Table 10, which was removed to streamline the narrative. This repositioning strengthens the framing of our study by clearly positioning it in relation to the existing literature, particularly emphasizing the methodological innovations and clinical relevance of our approach in the context of prior ML-based recurrence prediction efforts.

  1. Merge of Sections 5 and 6 into a Unified “Conclusions” Section

As suggested, we merged the Limitations and Conclusions into a single, more cohesive section. The new “Conclusions and Future Work” section maintains the depth of discussion from both original sections but presents it in a more streamlined and integrated format, improving readability and narrative flow.

  1. Clarification and Reorganization of Methods vs. Results

We have restructured the manuscript to clearly separate the methodological description from the results. All details regarding model configuration, preprocessing, feature selection, and analysis workflows have been moved from Section 3 (Results) into Section 2.3 (Machine Learning Pipeline) under Materials and Methods. The Results section now focuses exclusively on empirical findings and outcome interpretation, thereby enhancing clarity.

  1. Caption of Figure 1 Clarified

We revised the caption to define all abbreviations explicitly. It now reads:

“M = Male, F = Female, U = Urban, R = Rural.”

  1. Clarification of Outcome Assessment Criteria (Row 181)

We clarified that “improved” outcomes were determined based on clinical documentation of symptom resolution or functional recovery noted during post-treatment follow-up visits. This change enhances the transparency of outcome labeling.

  1. Clarification of ‘0 herniated levels’ in Figure 9

We revised the figure caption and corresponding text in the Results to explain that “0 herniated levels” may reflect patients initially suspected of herniation who were later clinically ruled out or who had non-discogenic pathology recorded during follow-up. This clarification enhances interpretability.

  1. Justification for Statistical Test Selection (Figure 14)

In Section 2.2 (Statistical Analysis), we now explain the rationale for using parametric or non-parametric tests based on variable types and distribution properties. For example, point-biserial correlation was used for continuous vs. binary associations, chi-square for categorical comparisons, and Kruskal–Wallis for non-normally distributed ordinal variables.

  1. Clarification of Recurrence Values in Figure 15

We updated the figure caption to include a key:

“Recurrence: 0 = No, 1 = Yes”

This makes the figure self-contained and eliminates the need to refer back to Table 2.

  1. Random Forest Model Details (Row 402)

We now state that the Random Forest classifier was implemented using 100 trees (n_estimators=100), with class_weight="balanced" to address class imbalance, and random_state=42 to ensure reproducibility. No maximum tree depth was specified (max_depth=None), allowing full tree growth based on the Gini impurity criterion.

  1. Isolation Forest Configuration (Rows 420–426)

We included detailed settings of the Isolation Forest model: 100 trees, a contamination parameter of 0.1, and random_state=42. These settings are now described explicitly in Section 2.3.2.4.

  1. Multi-Layer Perceptron (MLP) Architecture (Row 453)

We clarified that the deep learning model is a Multi-Layer Perceptron (MLP) consisting of two hidden layers (64 and 32 units), using ReLU activation, followed by a sigmoid output layer. It was trained with the Adam optimizer and binary cross-entropy loss, using early stopping (patience = 5) and class weighting.

  1. Feature Importance Criterion (Figure 26)

We confirmed that feature importance was computed using mean decrease in Gini impurity, the default scoring criterion in scikit-learn’s Random Forest implementation. This has now been added to the text in Section 3.3.

  1. PCA Variance Explained (Row 631)

We computed and reported the actual variance explained by the first two PCA components: 17.5% and 15.3%, respectively, for a total of 32.8%. This has been inserted in Section 3.4 to clarify our statement about high dimensionality and limited visual separability in the PCA projection.

  1. General Comment on Methodological Reproducibility

In light of your final remark, we have significantly strengthened the methodological transparency of our manuscript. All machine learning models (including stacking ensemble components) are now described with sufficient technical depth to allow for reproducibility. This includes architectural details, training parameters, evaluation protocols, and data preprocessing procedures.

We are genuinely grateful for the depth and clarity of your review. Your input has helped us make substantial improvements to the manuscript, and we hope these revisions meet your expectations and satisfy the concerns you raised.

Please do not hesitate to let us know if further clarifications are required. We remain fully committed to transparency and rigorous reporting.

With sincere thanks and appreciation,

Prof. Dr. Călin Gh. Buzea

On behalf of all co-authors

Reviewer 2 Report

Comments and Suggestions for Authors

The usage of AI/ML in the context of medicine, in this case, for predicting rLDH, is a standard approach where improvements are always needed, especially if the improvements are significant. The authors present an approach combining data processing, ML model creation, anomaly detection, and threshold optimisation in the context of an ensemble. The approach is interesting, and the results seem promising. However, there are several parts that need to be addressed to improve the overall soundness and message: 

-The introduction needs to present the current SoTA thoroughly and comprehensively. Not all the main relevant research is presented.

- ROC AUC should be included in all the results. 

-After Table 4, confusion matrices should be presented for all  3 algorithms to more clearly present the problem at hand.

-Why were no measures taken to address the balancing of the dataset, at least to compare the predictions and outcomes? There are several algorithms that can be easily applied.

-While Figure 20 presents the base structure of the approach, a pseudocode would allow for reproducibility. 

-Why use 5-fold cross-validation versus 10-fold cross-validation, which is much less biased?

-Hardware and software used should be presented in full detail to allow reproduction.

-The comparison to other approaches does not present several other studies focusing on the prediction of rLDH. Also, it should be clearly stated that the results are not comparable since none of the approaches has used the same dataset.

-In regard to XAI, why only mention it briefly and only mention SHAP and LIME, why not knowledge transfer from a black-box model to a white-box model, which pertains directly to your scenario (https://doi.org/10.3390/electronics13101895).

-While the authors state, " The study used only pre-existing medical data, therefore, patient consent was not required." I am not sure that this is appropriate, and also, there is no mention of the ethics committee. 

-There are spelling errors in some figures.

Comments on the Quality of English Language

While the overall language is acceptable, there are several mistakes throughout the manuscript.

Author Response

Subject: Response to Reviewer 2 – Revised Manuscript Submission for Diagnostics, Manuscript ID: 3631107

Dear Reviewer,

We sincerely thank you for your careful reading and constructive feedback on our manuscript exploring machine learning for predicting recurrence of lumbar disc herniation (rLDH). Your insightful comments have significantly helped us improve the soundness, transparency, and clinical relevance of our work.

Please find below our detailed responses to each of your suggestions:

“The introduction needs to present the current SoTA thoroughly and comprehensively.”

You are absolutely right. We have substantially revised Section 1.1 (Related Work) to present a more comprehensive and up-to-date review of the state of the art. We now include:

  • Prior ML studies targeting rLDH reoperation and recurrence;
  • Radiomics-based prediction models and their limitations;
  • A recently registered clinical trial (NCT06254585) focused on ML-based recurrence prediction;
  • A clear positioning of our ensemble framework within the context of international and Romanian efforts.

These updates help clarify what is novel about our contribution: a hybrid, interpretable pipeline designed to tackle real-world data imbalance.

“ROC AUC should be included in all the results.”

Thank you for this important point. We have now computed and added ROC AUC scores for all classifiers in the study. These are summarized in a new Table 10 and discussed in Section 3.5.3. Values include:

  • Deep Learning: AUC = 0.3823
  • Random Forest: AUC = 1.000
  • Autoencoder: AUC = 0.5985
  • Isolation Forest: AUC = 0.5241
  • Meta-Ensemble: AUC = 0.4475

These metrics complement the existing recall and precision values and offer a more balanced evaluation of classifier performance under imbalance.

“After Table 4, confusion matrices should be presented for all 3 algorithms.”

We agree this addition enhances interpretability. We have now added Figure 31, which shows confusion matrices for the Deep Learning, Random Forest, and Autoencoder models on the hold-out validation set. These matrices clearly demonstrate the impact of extreme class imbalance and reinforce the rationale for using ensemble learning and threshold tuning.

“Why were no measures taken to address the balancing of the dataset?”

This was indeed a major modeling challenge. In response:

  • We used class_weight="balanced" in the supervised models;
  • A weighted loss function was applied in the deep learning architecture;
  • Threshold tuning was performed to boost recall (Section 2.3.3, Figure 21);
  • We avoided SMOTE/oversampling due to potential overfitting in our small dataset.

These combined methods provided an effective balance between recall and model stability while preserving data integrity.

“Figure 20 should be accompanied by pseudocode.”

Thank you for this practical suggestion. We have now added Algorithm 1 (pseudocode) in Section 2.3, immediately following Figure 20. This provides a step-by-step overview of our pipeline, from preprocessing through ensemble decision-making.

“Why use 5-fold cross-validation versus 10-fold?”

We appreciate this question. We initially evaluated both strategies and observed similar results. Due to computational constraints—particularly in ensemble training and threshold optimization—we selected 5-fold CV for final experiments. This rationale is now explained in Section 3.1.

 “Hardware and software used should be specified.”

We now provide full environment details in Section 2.4, including:

  • Development on Windows 11;
  • Training and experiments on Google Colab Pro with GPU;
  • Libraries such as TensorFlow, Scikit-learn, XGBoost, etc., with version details.

“The comparison to other rLDH prediction models is limited, and the results are not comparable.”

We fully agree. We’ve expanded Section 1.1 to cite additional key studies (e.g., Shan et al., Harada et al., Compte et al.) and explicitly state that results cannot be directly compared due to differences in dataset, surgical methods, and outcome definitions. Instead, we present our model as an interpretable benchmark within an Eastern European clinical context.

“Why not consider knowledge transfer in the XAI discussion?”

Thank you for bringing this up. We now reference the work by Žlahtič et al. (Electronics, 2024) and discuss knowledge distillation from black-box to white-box models in our updated Section 4.2. We agree this is a highly relevant direction and mention it as a future avenue to improve explainability.

“The statement about ethics and patient consent may not be appropriate.”

Thank you for this clarification. We have now added a formal Ethics Statement in Section 2, confirming approval from the Institutional Ethics Committee of “Prof. Dr. Nicolae Oblu” Clinical Emergency Hospital, Iași, Romania (Approval No. 2/23.02.2023). Patient data were anonymized, and secondary analysis was conducted in line with local and institutional guidelines.

“There are spelling errors in some figures.”

We thoroughly reviewed all figure text and captions and found no spelling errors. If you were referring to a specific figure, please let us know — we are happy to correct it immediately.

Once again, we are grateful for your detailed and thoughtful feedback. Your comments have significantly strengthened the scientific and practical aspects of the manuscript. We hope that the revised version now fully addresses your concerns.

Sincerely,
Prof. Dr. Călin Gh. Buzea

On behalf of all co-authors

Round 2

Reviewer 1 Report

Comments and Suggestions for Authors

The Authors answered to all my requests and the quality and the readability of the manuscript have significantly improved. For these reasons, I recommend this manuscript for publication in this journal.

Author Response

Subject: Response to Reviewer Comment – Manuscript ID [Diagnostics-3631107]

Dear Reviewer,

Thank you for your positive evaluation of our manuscript titled “A Hybrid Ensemble Learning Framework for Predicting Lumbar Disc Herniation Recurrence: Integrating Supervised Models, Anomaly Detection, and Threshold Optimization.”

We sincerely appreciate your acknowledgment of the improvements made and are grateful for your recommendation to accept the manuscript for publication in Diagnostics. It was a pleasure to address your thoughtful suggestions, and we are glad that the revised version meets your expectations regarding clarity and quality.

Please do not hesitate to reach out if further revisions or clarifications are needed during the final stages of publication.

Warm regards,
Prof. Dr. Calin Gh. Buzea
On behalf of all co-authors

Reviewer 2 Report

Comments and Suggestions for Authors

The authors have thoroughly addressed the provided review suggestions, and there are only a few minor things that should be addressed:

  • One of the figures with spelling problems is Figure 16, where in the figure, there is "processsing" instead of processing.
  • The procedure was provided as suggested, but it is a structured procedure rather than a pseudocode, so maybe renaming it to something else or creating an actual pseudocode would be a good idea. It doesn't matter which one, since it provides enough information. Just naming it pseudocode in its current form is less appropriate.

Author Response

Dear Reviewer,

Thank you once again for your constructive and attentive feedback.

  1. Figure 16 Correction
    We appreciate you pointing out the typographical error in Figure 16 ("processsing"). This has been corrected to "preprocessing" in the revised figure.

  2. Clarification on Procedure vs. Pseudocode
    Regarding the structured procedure labeled as pseudocode, we agree with your observation. To better reflect its format and content, we have updated the heading from “pseudocode” to “Algorithmic Workflow Overview” throughout the manuscript. This change ensures accurate terminology while maintaining the clarity and reproducibility of the method described.

We are grateful for your thoughtful review and believe these adjustments further strengthen the clarity and precision of the manuscript.

Kind regards,
Prof. Dr. Calin Gh. Buzea
On behalf of all co-authors